# Lysosomal damage drives mitochondrial proteome remodelling and reprograms macrophage immunometabolism

Claudio Bussi [1] ✉, Tiaan Heunis [2,6], Enrica Pellegrino[1], Elliott M. Bernard [1,7], Nourdine Bah[1], Mariana Silva Dos Santos [1], Pierre Santucci[1,8], Beren Aylan[1], Angela Rodgers[1], Antony Fearns[1], Julia Mitschke[3,4], Christopher Moore[1], James I. MacRae [1], Maria Greco[1,9], Thomas Reinheckel [3,4,5], Matthias Trost[2] & Maximiliano G. Gutierrez [1] ✉

Transient lysosomal damage after infection with cytosolic pathogens or silica crystals uptake results in protease leakage. Whether limited leakage of lysosomal contents into the cytosol affects the function of cytoplasmic organelles is unknown. Here, we show that sterile and non-sterile lysosomal damage triggers a cell death independent proteolytic remodelling of the mitochondrial proteome in macrophages. Mitochondrial metabolic reprogramming required leakage of lysosomal cathepsins and was independent of mitophagy, mito-proteases and proteasome degradation. In an in vivo mouse model of endomembrane damage, live lung macrophages that internalised crystals displayed impaired mitochondrial function. Single-cell RNA-sequencing revealed that lysosomal damage skewed metabolic and immune responses in alveolar macrophages subsets with increased lysosomal content. Functionally, drug modulation of macrophage metabolism impacted host responses to *Mycobacterium tuberculosis* infection in an endomembrane damage dependent way. This work uncovers an inter-organelle communication pathway, providing a general mechanism by which macrophages undergo mitochondrial metabolic reprograming after endomembrane damage.

The permeabilization and rupture of lysosomes after stress is a biological process relevant to immunity, neurodegeneration and cancer. To cope with the detrimental effects of lysosomal damage, cells possess mechanisms to either degrade or repair damaged endomembranes[1,2]. If unchecked, lysosomal damage and leakage of proteases into the cytosol will trigger inflammation and ultimately cell death[3,4]. There is however compelling evidence that endomembrane damage[5] or some continuous inflammatory states[6] do not always result in cell death and important cellular functions are actually regulated by limited lysosomal damage such as antigen presentation or chromatin remodelling[7–9]. The existence of multiple endomembrane repair mechanisms suggests that transient cytosolic leakage of luminal contents occurs, but the damage is efficiently resolved with cells returning to homeostasis. However, it is unknown if this transient leakage of

[1]The Francis Crick Institute, London, UK. [2]Biosciences Institute, Newcastle University, Newcastle, UK. [3]Institute for Molecular Medicine and Cell Research, Medical Faculty, Albert-Ludwigs-University Freiburg, Freiburg, Germany. [4]German Cancer Consortium (DKTK), German Cancer Research Center (DKFZ), Heidelberg, Germany. [5]Signalling Research Centres BIOSS and CIBSS, Albert-Ludwigs-University Freiburg, Freiburg, Germany. [6]Present address: Sir William Dunn School of Pathology, University of Oxford, Oxford, UK. [7]Present address: Department of Biochemistry, University of Lausanne, Epalinges, Switzerland. [8]Present address: Aix-Marseille Univ, CNRS, LISM, IMM FR3479, Marseille, France. [9]Present address: Radcliffe Department of Medicine, University of Oxford, Oxford, UK. ✉e-mail: claudio.bussi@crick.ac.uk; max.g@crick.ac.uk

lysosomal contents will impact the function of organelles in proximity of the endomembrane damage in homeostatic conditions. In this context, if there are cell death-independent pathways by which rupture of lysosomes regulates organelle function is poorly understood.

Mitochondrial metabolic reprogramming is critical for the immune function of myelocytic cells[10]. Whereas many studies have highlighted functional interactions between mitochondria and lysosomes in non-myelocytic cells, little is known about the mechanisms by which mitochondrial interactions with lysosomes regulate the immunometabolic activities of macrophages. This is important when we consider that the lysosomal composition is different in cells of the myelocytic lineage[11].

Mitochondria communicate via vesicular-mediated trafficking and contact with other organelles to regulate cellular metabolism and homeostasis. In particular, the crosstalk between endolysosomes and mitochondria is central for their metabolic function[12,13]. In addition to selective ubiquitin-mediated degradation of mitochondrial proteins by the proteasome, endolysosomal degradation pathways contribute to preserve mitochondria integrity[14–16]. After localised mitochondrial oxidative stress, the budding of mitochondrial-derived vesicles (MDV) delivers specific mitochondrial contents to endolysosomes for degradation[17]. When the mitochondrial damage is extensive, mitochondria are removed by mitophagy and targeted to lysosomes[14–16,18]. If this continuous cross talk between lysosomes and mitochondria is affected during lysosomal damage is unknown.

Here, we combined the use of human iPSC-derived macrophages (iPSDM) with an in vivo mouse model of endomembrane damage to test the hypothesis that transient protease leakage, induced by limited lysosomal damage, has an effect in neighbouring organelles. We found that leaked lysosomal proteases caused mitochondrial protein degradation and impaired mitochondrial function with changes that result in altered macrophage metabolic and immune responses. Our data uncovered an unexpected interorganelle interaction mechanism that regulates homeostatic immune responses to infection.

## Results

### Lysosomal protease leakage triggers mitochondrial protein degradation in macrophages

Given that lysosomes interact with mitochondria, we tested if lysosomal damage had an effect on mitochondria in a human iPSC-derived macrophage (iPSDM) model[19]. We triggered sub-lethal membrane damage in human iPSDM with the lysosomotropic agent L-leucyl-L-leucine methyl ester (LLOMe), infection with the intracellular pathogen *Mycobacterium tuberculosis* (Mtb) and phagocytosis of silica crystals (Fig. 1a and Supplementary Fig. 1a–c). The three endomembrane damaging agents consistently decreased the levels of the outer mitochondrial membrane (OMM) proteins TOM20 and Mitofusin 2 (MFN2) and the inner mitochondrial membrane protein (IMM) TIM23 (Fig. 1b) without affecting cell viability in any of the conditions tested (Supplementary Fig. 1e–g). Conversely, the levels of selected matrix proteins Heat-shock protein-60 (HSP-60) and Citrate synthase (CS) remained unchanged (Fig. 1b). The reduction in OMM and IMM proteins was dependent on proteolytic activity since a protease inhibitor (PI) cocktail, that includes many typical lysosomal protease inhibitors such as E64 (cysteine-type cathepsins) and Pepstatin (aspartic-type cathepsins), and the cathepsin B selective inhibitor CA074-Me inhibited the degradation of these proteins (Fig. 1b). Neither phagocytosis of silica beads[20] nor infection with Mtb ΔRD1, a mutant strain unable to induce endolysosomal damage[21], had an effect on the mitochondrial protein levels, indicating this mitochondrial protein degradation required endomembrane damage (Fig. 1b). This effect was independent of proteasome activity since the proteasome inhibitor bortezomib (BTZ) did not prevent mitochondrial protein degradation (Fig. 1b). These observations were confirmed in primary human monocyte-derived macrophages (hMDM, Supplementary Fig. 1h). In contrast, LLOMe treatment in cell lines such as HeLa, HEK293T or iPSC used for macrophage differentiation, did not reduce mitochondrial protein levels even at higher concentrations and longer exposure (Supplementary Fig. 1i). Treatment with lysosomal protease or proteasome inhibitors alone did not change the mitochondrial protein levels (Supplementary Fig. 1j). Notably, the total mass of mitochondria at the single macrophage level remained unaltered or slightly decreased in the conditions tested, suggesting changes in mitochondrial composition rather than in total mitochondria numbers (Supplementary Fig. 1k). To define if the endomembrane-damage dependent mitochondrial protein degradation was dependent on autophagy and/or proteasome degradation, ATG7 and Parkin (PRKN) respectively were knocked out to generate single- and double- knockout (KO) iPSDM (Fig. 1c and Supplementary Fig. 1l). Consistent with our previous results, LLOMe-induced endomembrane damage in the single- or double- KO (DKO) iPSDM decreased OMM and IMM protein levels that were rescued in the presence of a PI cocktail but not with BTZ (Fig. 1c). To confirm the role of lysosomal proteases in this process, membrane damage was induced with LLOMe in bone marrow derived macrophages (BMDM) from mice knocked out for Cathepsin B (CtsB), Cathepsin L (CtsL) or Cathepsin S (CtsS) that are highly expressed in macrophages[11]. In BMDM lacking CtsB or CtsL the degradation of mitochondrial proteins was significantly reduced, indicating that both CtsB and CtsL contribute to mitochondrial protein degradation after endolysosome damage (Fig. 1d and Supplementary Fig. 1d). In addition, iPSDM expressing the mitophagy construct NIPSNAP[18] did not show a significant increase in m-Cherry only puncta (indicative of mitophagy) after Mtb infection, LLOMe or silica crystals treatment (Fig. 1e, f). Moreover, confocal imaging and ultrastructural analysis of mitochondria in iPSDM, did not show evidence of increased mitochondrial-derived vesicles (MDVs) in any condition tested[17] (Fig. 1g, h and Supplementary Fig. 1m). We concluded that lysosomal damage induces proteolytic degradation of mitochondrial proteins primarily in macrophages independently of mitophagy and proteasome.

### Lysosomal leakage induces mitochondrial proteome remodelling

Mitochondria have been isolated from many cell types but to our knowledge, the mitochondrial proteome in human macrophages has not been characterised. To further define the consequences of endolysosomal leakage for the mitochondria in human macrophages, we generated human iPSC stably expressing MITO-tag[22] (Fig. 2a and Supplementary Fig. 2). The MITO-tag iPSCs were differentiated into iPSDM and treated or not with LLOMe followed by mitochondria immunoprecipitation and label-free proteomics analysis (Fig. 2a and Supplementary Fig. 2a–d). Further quality control analysis of the MITO-tag proteomics data after median normalization revealed distinct clustering of treatment groups when using Pearson correlation coefficients or when performing principal component analysis (PCA, Supplementary Fig. 2e–h). PCA analysis showed that the mitochondrial proteome of LLOMe-treated cells in the presence of protease inhibitors clustered closer to untreated than LLOMe-only treated cells (Supplementary Fig. 2g, h). Confirming our previous observations, we observed a significant reduction of total mitochondrial proteins after LLOMe treatment (Fig. 2b). Unlike our WB analysis, by covering the whole mitochondrial proteome, we found a decrease of not only OMM and IMM proteins but also matrix proteins (Fig. 2b, Supplementary Table 1 and Supplementary Data 1). Notably, we observed a predominant decrease of IMM proteins belonging to the electron transport chain (ETC) in LLOMe-stimulated macrophages (Supplementary Table 1 and Supplementary Data 1) and gene ontology (GO) enrichment analysis confirmed this pathway was altered after membrane damage (Fig. 2c, Supplementary Table 1 and Supplementary Data 1). The reduction in IMM and ETC proteins was reverted with a PI cocktail

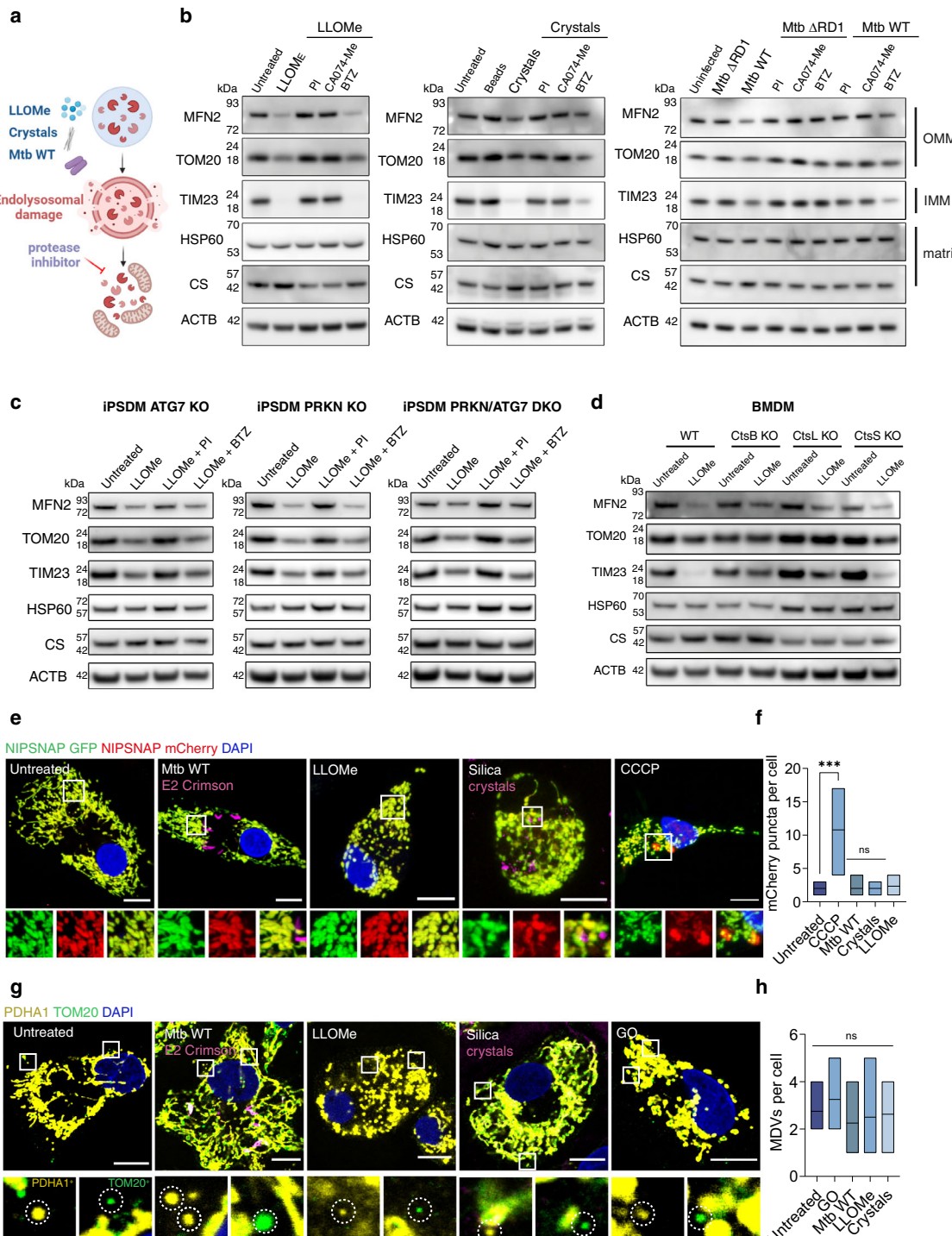

**Fig. 1 | Lysosomal protease leakage triggers mitochondrial protein degradation in macrophages. a** Schematic showing the conditions tested in this study. **b** Immunoblots for MFN2, TOM20, TIM23, HSP60 and Citrate synthase (CS) in iPSDM stimulated with 0.5 mM LLOMe for 1 h, 100 µg/mL silica crystals or beads for 3 h or infected with Mtb WT or Mtb ΔRD1 for 48 h and incubated with the indicated protease or proteasome inhibitors. Beta-actin (ACTB) levels were used as loading controls (repeated three times with similar results). **c** Immunoblots for mitochondrial proteins in iPSDM WT, ATG7 KO, PRKN KO and PRKN/ATG7 DKO stimulated with 0.5 mM LLOMe for 1 h and incubated in the presence or absence of the indicated inhibitors (repeated three times with similar results). **d** Immunoblots for mitochondrial proteins in BMM WT, CtsB KO, CtsL KO and CtsS KO stimulated with 0.5 mM LLOMe for 1 h. (repeated three times with similar results). **e** Representative images of iPSDM expressing the mitophagy reporter NIPSNAP and stimulated with 0.5 mM LLOMe for 1 h, 100 µg/mL silica crystals for 3 h, infected with Mtb WT for 48 h or treated with 20 µM CCCP for 3 h. **f** NIPSNAP mCherry only puncta evaluated by confocal microscopy, $n = 30$ cells examined per condition over three independent experiments. **g** TOM20+/PDHA1− and PDH+/TOM20− MDVs were quantified after the indicated conditions. Glucose oxidase (GO) was used at 50 mU/ml for 1 h as a positive control, $n = 20$ cells examined per condition over three independent experiments. **h** One-way ANOVA and Tukey post-test was used for multiple comparisons ***$p \leq 0.001$, ns no significant. Scale bars, 10 µm. Floating bar plots show minimum and maximum values, line at mean. Unprocessed blots and Source data are provided as a Source Data file.

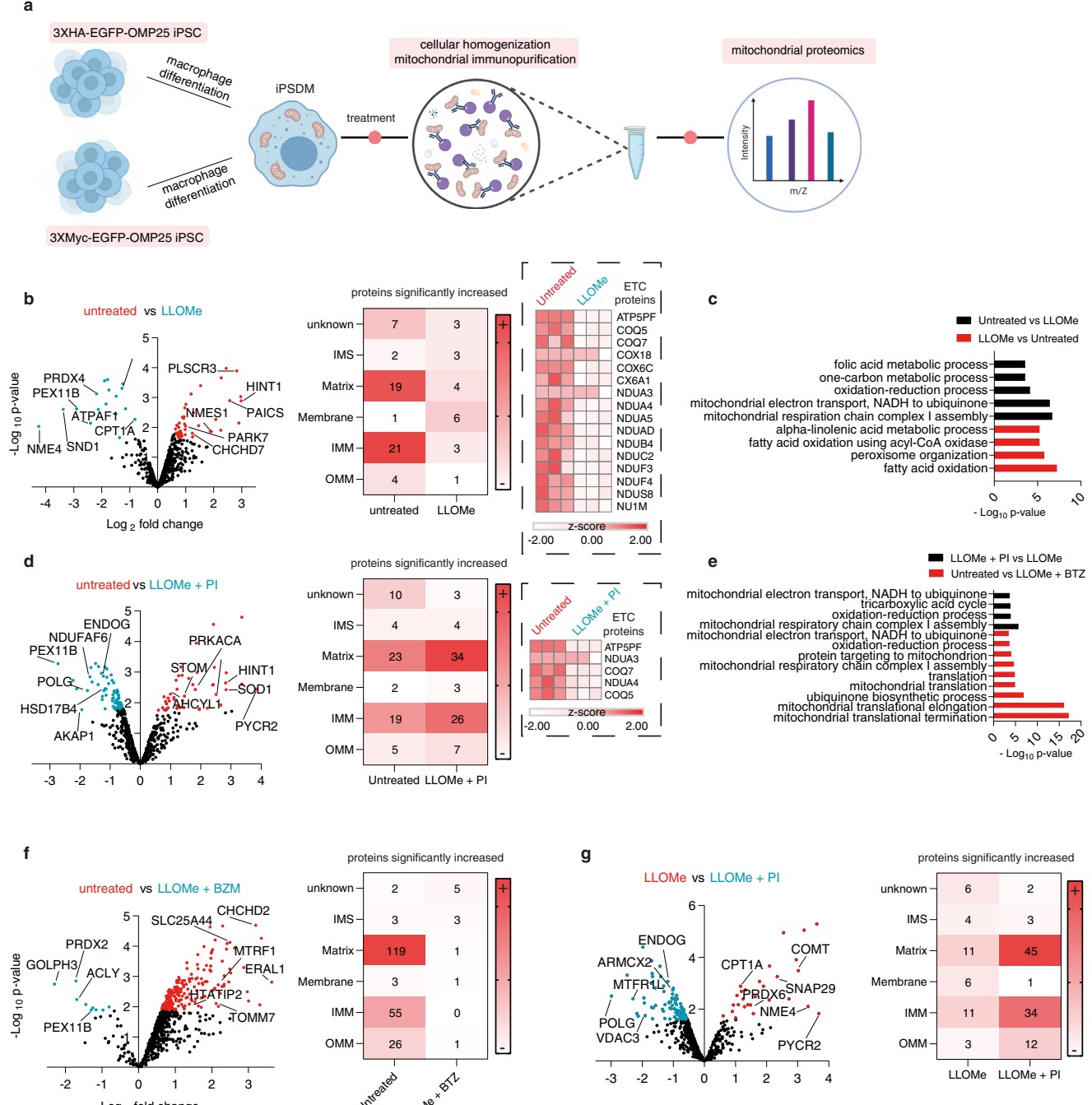

**Fig. 2 | Lysosomal leakage remodels the mitochondrial proteome in macrophages. a** Schematic showing the MITO-tag workflow in iPSDM. **b** Proteomic analysis of isolated mitochondria from MITO-tag iPSDM untreated (red) or treated with LLOMe (blue). Volcano plot shows proteins with fold change > 1.5 and an adjusted *p*-value ≤ 0.05. The two most upregulated proteins in the OMM, IMM and matrix (Supplementary Data 1) are shown as illustrative example. Heat maps show the mitochondrial localization of the proteins that were significantly increased in the indicated comparison and ETC proteins significantly decreased after LLOMe treatment are shown (dashed box). Annotations were obtained from MitoCarta3.0. **c** Significantly regulated proteins from the indicated conditions (Supplementary Table 1) were used for Gene Ontology (GO) cellular component enrichment analysis using the Gene Functional Annotation Tool available at the DAVID v6.7 website

(https://david.ncifcrf.gov/). A maximum *p*-value of 0.05 (Benjamini) was chosen to select only significantly enriched GO cellular components. **d** Volcano plot and mitochondrial protein localization as in (**a**) in the presence or absence of protease inhibitors (PI). The ETC proteins from (**a**) significantly decreased in the presence of the PI are shown (dashed box). See also Supplementary Table 2. **e** Bar graph showing GO cellular component enrichment analysis from the indicated conditions as in (**c**). **f** Volcano plot and mitochondrial protein localization as in (**a**) in the presence or absence of the proteasome inhibitor Bortezomib (BTZ). **g** Volcano plot and mitochondrial protein localization as in (**a**) comparing LLOMe-treated iPSDM in the presence or absence of PI. *n* = 3 independent experiments. Source data are provided in Supplementary Data 1.

but not with BTZ (Fig. 2d–f, Supplementary Table 1 and Supplementary Data 1), indicating that this effect was dependent on protease but not proteasome activity. In agreement with these observations, LLOMe-treated macrophages in the presence of the PI cocktail showed a similar proteomic and GO enrichment profile to the untreated macrophages when compared to LLOMe stimulation alone (Fig. 2f, g). Although endomembrane damage induced an overall decrease of mitochondrial proteins, we observed that those associated with fatty acid beta-oxidation (FAO) and peroxisomes were significantly upregulated in LLOMe-treated macrophages (Fig. 2c, Supplementary Table 1 and Supplementary Data 1). Altogether, lysosomal damage in macrophages induced a broad protease-dependent degradation of the mitochondrial proteome with a decrease in ETC- and an increase in FAO-associated proteins.

## Lysosomal leakage impacts mitochondrial function

Next, using single-cell high-content analysis (Supplementary Fig. 3a and b), we investigated if these changes in the mitochondrial proteome after lysosomal damage had an impact on mitochondrial function. Only conditions that induced lysosomal damage resulted in a decreased intensity of the mitochondrial membrane potential sensor iTMRM, indicating a reduction in the number of active mitochondria (Fig. 3a). This lysosomal damage-dependent reduction in mitochondrial membrane potential was rescued in presence of a PI cocktail arguing that proteases leaking out of lysosomes impacted mitochondrial activity (Fig. 3b). Consistent with our previous results, the reduction of the mitochondrial membrane potential after inducing lysosomal damage was dependent on proteolytic activity and independent of ATG7 and PRKN (Supplementary Fig. 3c). Mitochondrial turnover measured with the oxidation-dependent green-to-red photoconvertible reporter MitoTimer[23] (Supplementary Fig. 3d) showed that LLOMe, silica crystals or infection with Mtb WT significantly accelerated mitochondrial maturation (Fig. 3c, d). Moreover, macrophages expressing the mitochondrial hydrogen peroxide ($H_2O_2$) sensor pHyPer-dMito[24] (Supplementary Fig. 3e) showed an increase in mitochondrial $H_2O_2$ levels, one of the main reactive oxygen species (ROS) formed by mitochondria and generated by cells[25], after endomembrane damage (Fig. 3e, f). In agreement with our previous results, the use of a protease inhibitor rescued the increased values observed with these functional probes after inducing lysosomal damage (Supplementary Fig. 3f). Confirming a role for cathepsins in the lysosomal damage-dependent impact on mitochondrial function, the mitochondrial membrane potential was higher in BMDM CtsB KO and CtsL KO when compared to LLOMe-treated BMDM WT. In contrast, no differences were observed in BMDM CtsS KO (Fig. 3g, h). Pre-treating iPSDM with the V-ATPase inhibitor Bafilomycin A1 (BAFA1) impaired the protease leakage-dependent mitochondrial protein and activity decrease (Supplementary Fig. 3g, h). Next, we transiently expressed the cytosolic inhibitor of cysteine cathepsins Cystatin B tagged with GFP (CSTB-GFP)[26]. The expression of CSTB-GFP in the cytosol reverted the reduction in the number of active mitochondria after Mtb infection and LLOMe or silica crystals treatment as measured by iTMRM intensity (Fig. 3i, j). Altogether, lysosomal damage caused by three different stimuli induced a reduction in macrophage mitochondrial function that primarily requires CtsB and CtsL activity.

## Mitochondrial activity is affected by proximity to damaged lysosomes

Because the observed effect requires lysosomal damage and cathepsins, we addressed how leaked proteases target the mitochondria and if proximity between lysosomes and mitochondria was required. By live-cell super resolution imaging in silica crystals or LLOMe-treated iPSDM, we observed that mitochondria with reduced membrane potential were in the proximity of Gal3-positive vesicles (Fig. 3k, l and Supplementary Fig. 4a, b), indicating that lysosomal damage can locally impact mitochondrial activity. These live cell observations were further confirmed by high-content single-cell analysis, where Mito-Tracker Deep Red intensity was reduced in the proximity of Gal-3 positive damaged endolysosomes (Fig. 3m, n). To further understand the nature of these interactions, we pre-loaded endolysosomes with 5 nm gold, then induced damage with LLOMe and analysed if gold particles could be found in the mitochondria. In contrast to control conditions where gold particles were only present in endocytic organelles, LLOMe-treated macrophages showed gold particles throughout the cytosol and inside mitochondria (Supplementary Fig. 4c, d) suggesting that lysosomal contents are leaked into the mitochondria after damage. In agreement with these results, we detected a significant enrichment of lysosomal cathepsins in the mitochondrial proteome of macrophages treated with LLOMe and not in the untreated controls, excluding a potential contamination from the mitochondrial immunopurification (Supplementary Fig. 4e).

A recent study showed that viable cells under oxidative stress conditions form pores by the voltage-dependent anion channel (VDAC) oligomers in the mitochondrial outer membrane[27]. In agreement with our previous results showing increased mitochondrial ROS after lysosomal damage, we found that LLOMe induced VDAC oligomerisation (Supplementary Fig. 4f), while we did not find changes in the total protein levels in our mitoproteomics analysis (Supplementary Fig. 4g). Moreover, the use of the VDAC1 oligomerisation inhibitor VBIT-4[28] but not the BAX oligomerisation inhibitor BAI[29] rescued the IMM but not the OMM protein levels decrease observed after endolysosomal damage (Supplementary Fig. 4h). In addition, pre-treatment with VBIT-4 also reverted the mitochondrial activity impairment after LLOMe treatment (Supplementary Fig. 4i). Altogether these results suggest a scenario where the proteolytic activity of lysosomal leaked proteases requires organelle proximity, and it might be further enhanced by mitochondrial porins-mediated lysosomal transfer.

We next tested if mitoproteases[30] could also mediate the mitochondrial quality impairment observed after lysosomal damage. In agreement with previous evidence[11], macrophages displayed increased lysosomal activity in comparison with iPSC, HeLa or HEK cells. (Supplementary Fig. 5a–c). Intriguingly, LLOMe-induced limited lysosomal damage reduced mitochondrial activity, as measured by iTMRM, in iPSDM but not in these other cell types (Supplementary Fig. 5d). Unlike lysosomal cathepsins, we did not observe in our mitochondrial proteomic analysis an enrichment in mitoproteases after LLOMe treatment (Fig. 4c and Supplementary Fig. 5e). In addition, we did not find the mitochondrial proteases YME1L1, LONP1 or endopeptidase ClP, a subunit of the CLPXP protease, selectively enriched in macrophages, which also differs from what we observed for lysosomal proteases (Supplementary Fig. 5f–h). Pre-treating iPSDM with inhibitors of mitoproteases 1,10-phenanthroline (o-Phe)[31,32], TPEN[33] and A2-32-01[34,35] did not revert the decrease of mitochondrial protein levels or activity after endolysosomal damage (Supplementary Fig. 5i, j), ruling out mitoproteases as mediators of the mitochondrial quality impairment after endomembrane damage.

Because we found that proximity between the damaged lysosome and mitochondria impacts mitochondrial activity, we tested the role of mitochondria–lysosome (M-L) contacts[12]. The M-L contacts measured with the lysosomal activity based-probe[36] revealed that duration of the M-L contact had a dependency on the lysosome size (Fig. 4a–e, Supplementary Movie 1–2). There was no difference in the percentage of M-L contacts when considered a minimum duration of 20 s (Fig. 4c). The overexpression of EGFP-Rab7 or the constitutively active mutant EGFP-Rab7Q67 induced endolysosome enlargement (Supplementary Fig. 6a, b), extending the duration of M-L contacts (Fig. 4e) up to 10 min. Similar results were observed when overexpressing LAMP-1

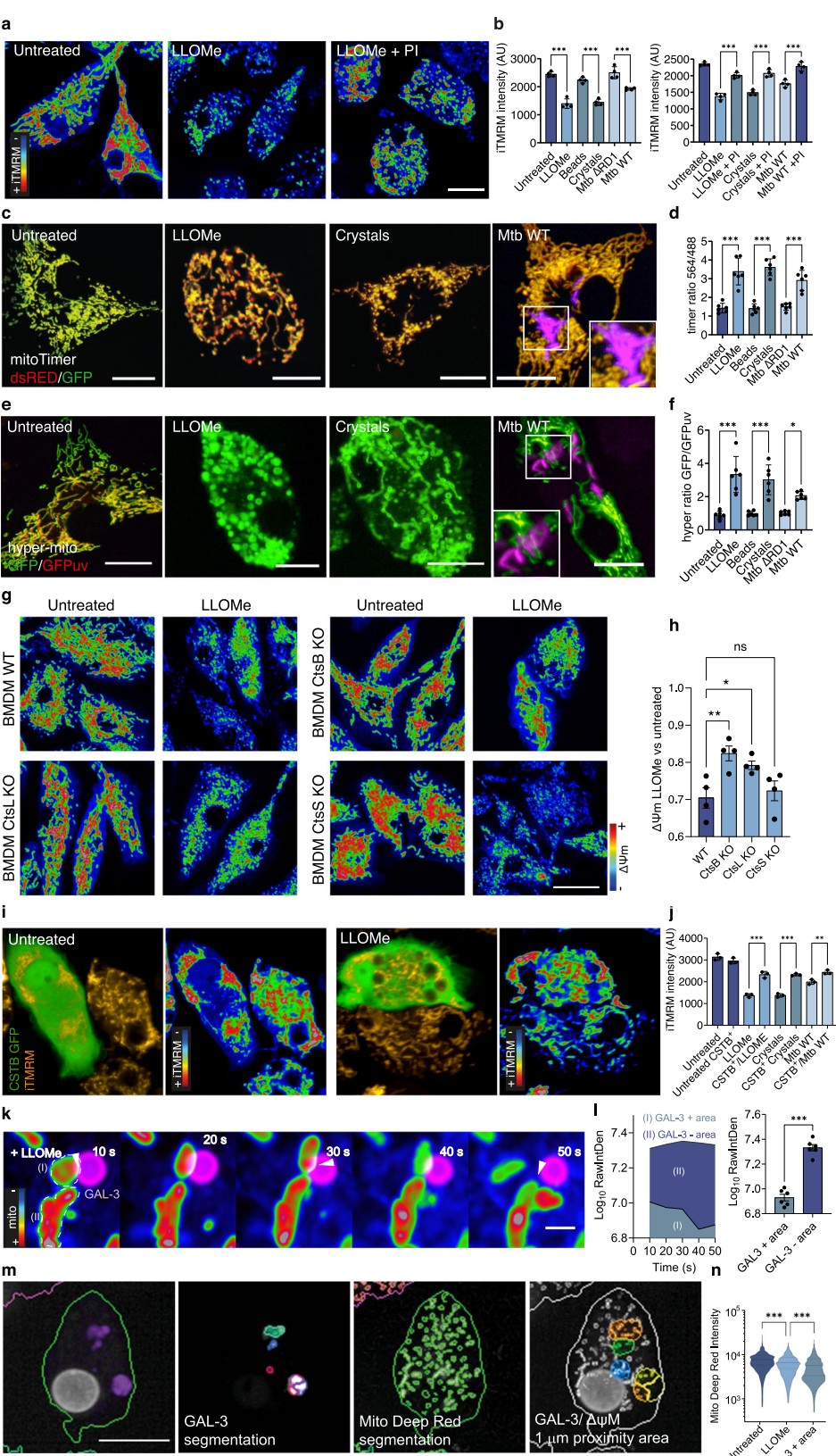

(Fig. 4e and Supplementary Fig. 6c), indicating that the enlargement of lysosomes increased the duration of M-L contacts. We found that the decrease in mitochondrial activity after endomembrane damage occurs at earlier time points in macrophages expressing EGFP-Rab7, EGFP-Rab7Q67 or LAMP-1-mNeonGreen (Fig. 4f–h). Intriguingly, similar results were observed when mitochondrial activity was tracked

intracellularly considering the size of the lysosomes with which the mitochondria interact (Fig. 4i–l and Supplementary Movie 3–5). In line with our previous observations correlating lysosomal size, duration of M-L contact and mitochondrial activity, we observed that, after inducing lysosomal damage, the mitochondrial membrane potential decrease was faster in the intracellular regions where the mitochondria

**Fig. 3 | Lysosomal leakage impacts mitochondrial activity in macrophages.**
**a** iTMRM intensity levels in iPSDM untreated or treated with 0.5 mM LLOMe in the presence or absence of PI. **b** Quantification of iTMRM intensity in iPSDM treated with LLOMe for 1 h, 100 μg/mL silica crystals or beads for 3 h or infected with Mtb WT or Mtb ΔRD1 for 48 h and incubated in the presence or absence of PI. **c**, **d** iPSDM expressing mito-timer (**c**) and fluorescence ratio evaluation by high-content imaging followed by mitochondrial segmentation (**d**). **e**, **f** iPSDM expressing hypermito (**e**) and fluorescence ratio quantification (**f**). **g**, **h** iTMRM intensity levels (**g**) and quantification (**h**) of BMM WT, CtsB KO, CtsL KO and CtsS KO stimulated with 0.5 mM LLOMe for 1 h (**i**) iTMRM intensity levels in iPSDM expressing or not CSTB C-GFPSpark and untreated or not with LLOMe. **j** Quantification of iTMRM intensity in iPSDM expressing CSTB C-GFPSpark and treated as in (**b**). One-way ANOVA and Tukey post-test was used for multiple comparisons. **k** iPSDM expressing GAL-3-RFP and incubated with MitoTracker Deep Red were treated with 0.5 mM of LLOMe and

imaged immediately after stimulation at 1 frame per 10 s. A selected sequence showing a GAL-3 positive vesicle in proximity of mitochondria is shown. Scale bar: 1 μm. **l** MitoTracker Deep Red intensity quantification of mitochondrial areas in proximity of GAL-3- positive vesicles or without interaction (GAL-3-negative), illustrated as "I" and "II", respectively. Bar plots show 12 events per condition from one out of three representative experiments. A paired *t*-test test was used for comparisons. Images shown are z-stack projections. **m**, **n** High-content single-cell analysis pipeline (**m**) and mitochondrial membrane potential quantification (**n**) of iPSDM expressing GAL-3-RFP. Mitochondrial regions around 1 μm distance from a GAL-3 positive vesicle are considered as GAL-3 positive areas. *$p \le 0.05$; **$p \le 0.01$; ***$p \le 0.001$. Images shown are z-stack projections. Bar plots show data mean values +/− SEM of at least three independent experiments. Scale bars: 10 μm. Source data are provided as a Source Data file.

were in contact with enlarged lysosomes (Fig. 4i–l and Supplementary Movie 3–5). Altogether, these data indicate that the mitochondrial activity and proteome are primarily affected by leaked proteases in the proximity of mitochondria.

## Lysosomal leakage reprograms macrophage metabolism

We next investigated if these changes in mitochondrial activity after lysosome damage had an effect in macrophage metabolism. Extracellular flux analysis (EFA) showed that resting iPSDM were not polarized towards a glycolytic or oxidative phosphorylation (OXPHOS) phenotype presenting relatively low values of oxygen consumption rate (OCR, indicative of OXPHOS) and extracellular acidification rate (ECAR, indicative of glycolysis) (Fig. 5a–c). In contrast, treatment with LLOMe induced an increase in the levels of OCR and ECAR (Fig. 5a). This effect was reverted when iPSDM were concomitantly treated with PI cocktail and the solely addition of the inhibitor had no effect (Fig. 5a). Infection of iPSDM with Mtb or treatment with silica crystals induced a similar metabolic phenotype that was dependent on the ability of inducing endomembrane damage since infection with Mtb ΔRD1 or treatment with silica beads slightly changed the metabolic profile. The metabolic changes induced after Mtb WT or silica crystals were also dependent on proteolytic activity (Fig. 5a–c). Despite the similar basal metabolic profile among the conditions studied, different responses were observed after challenging iPSDM with mitochondrial modulators such as oligomycin, Carbonyl cyanide-4 (trifluoromethoxy) phenylhydrazone (FCCP) and rotenone/antimycin (Fig. 5a–c). In line with an increased ATP demand, the ATP-linked respiration was higher after lysosomal damage, and this effect was also rescued in the presence of a protease inhibitor. On the other hand, only Mtb WT infection increased the spare respiratory capacity, while LLOMe and silica crystals induced a decrease in this parameter, suggesting different levels of mitochondrial impairment after lysosomal damage (Fig. 5a–c). To further evaluate the metabolic dynamics of iPSDM, we conducted EFA at a later time point after inducing lysosomal damage. Notably, we observed a significant protease-dependent decrease in the OCR, ATP-linked respiration, and spare respiratory capacity levels after LLOMe or silica crystals treatment (Fig. 5d). In agreement with this metabolic dynamic and with a lysosomal protease-dependent phenotype, we observed similar results in BMDM treated with LLOMe where CtsB and CtsL KO BMDM exhibited a less pronounced metabolic polarization (Supplementary Fig. 7a) and maintained mitochondrial integrity even at a later stage after endomembrane damage (Supplementary Fig. 7b). These results suggest that the apparent increased OCR levels observed immediately after lysosomal damage might be driven by elevated mitochondrial ROS production (Fig. 3e, f) rather than by an upregulation of OXPHOS activity[37,38], which impairment is more evident when macrophages are evaluated at a later time point after inducing endomembrane damage. In accordance with this hypothesis, the use of the mitochondrion-specific superoxide scavenger Mito-TEMPO[39] reduced the OCR but not the ECAR basal levels after LLOMe treatment (Supplementary Fig. 7c).

To define the global metabolic response to endomembrane damage in iPSDM, we conducted liquid chromatography (LC)–mass spectrometry (MS)-based metabolite profiling. Consistent with the EFA analysis showing increased ECAR levels in iPSDM after endomembrane damage, a metabolic pathway analysis of ¹³C-labelled metabolites revealed a predominant increase in the glycolytic flux. There was a marked elevation of lactate levels in LLOMe-treated iPSDM (Fig. 5e and Supplementary Fig. 7d) and this metabolic response required proteolytic activity (Fig. 5e). Notably, a lipidomics profile of macrophages undergoing lysosomal damage revealed an accumulation of lipids associated with endomembrane damage such as ceramides and lysophospholipids[40] (Supplementary Fig. 7e and Supplementary Data 2). By conducting NanoString gene expression analysis, we did not observe a significant metabolic transcriptional response after 1 h of LLOMe treatment (Supplementary Fig. 7f). However, silica crystals uptake or Mtb WT but not Mtb ΔRD1 infection triggered a down-regulation of mitochondrial respiration transcripts and an upregulation of transcripts associated with the HIF/glycolytic pathway (Supplementary Fig. 7g, h). Thus, protease leakage after lysosomal damage results in macrophage metabolism reprogramming through an enhanced glycolytic metabolism and impaired mitochondrial OXPHOS.

## Lysosomal protease leakage induces a decrease of mitochondrial activity and protein levels in vivo

To validate these macrophage responses to endomembrane damage in vivo, we used a mouse model of silica-induced damage[4]. Mice were treated with intratracheal instillation of either silica beads or silica crystals and after 18 h, bronchoalveolar lavage (BAL) samples were collected and analysed (Fig. 6a). Under these conditions, cells that internalised beads or silica crystals were viable (Fig. 6b). Notably, mitochondrial membrane potential at the single cell level was significantly lower in F4/80+ macrophages that phagocytosed crystals when compared to macrophages that phagocytosed beads or did not internalise either beads or crystals (Fig. 6c, f). Phagocytosis of silica crystals but not silica beads by F4/80⁺ lung macrophages resulted in membrane damage recognition by Gal-3 (Fig. 6d–f). By 3D imaging analysis of F4/80+ macrophages from the BAL, we found that Gal-3-positive vesicles containing silica crystals were in the proximity of mitochondria with reduced membrane potential (Fig. 6e). In agreement with the previous results using human macrophages, the protein levels of MFN2, TOM20 and TIM23 were significantly reduced in macrophages that internalised silica crystals when compared to silica beads (Fig. 6g). Consistent with our functional studies in human macrophages, phagocytosis of silica crystals in vivo increased oxygen consumption levels, likely produced by elevated ROS production (Fig. 6e, f and Supplementary Fig. 7c), and ECAR values, rendering the macrophages into a glycolytic profile (Fig. 6h). Thus, in a mouse model of in vivo endomembrane damage, macrophages that phagocytosed silica crystals have a marked reduction in mitochondrial activity and switched to a glycolytic metabolic state.

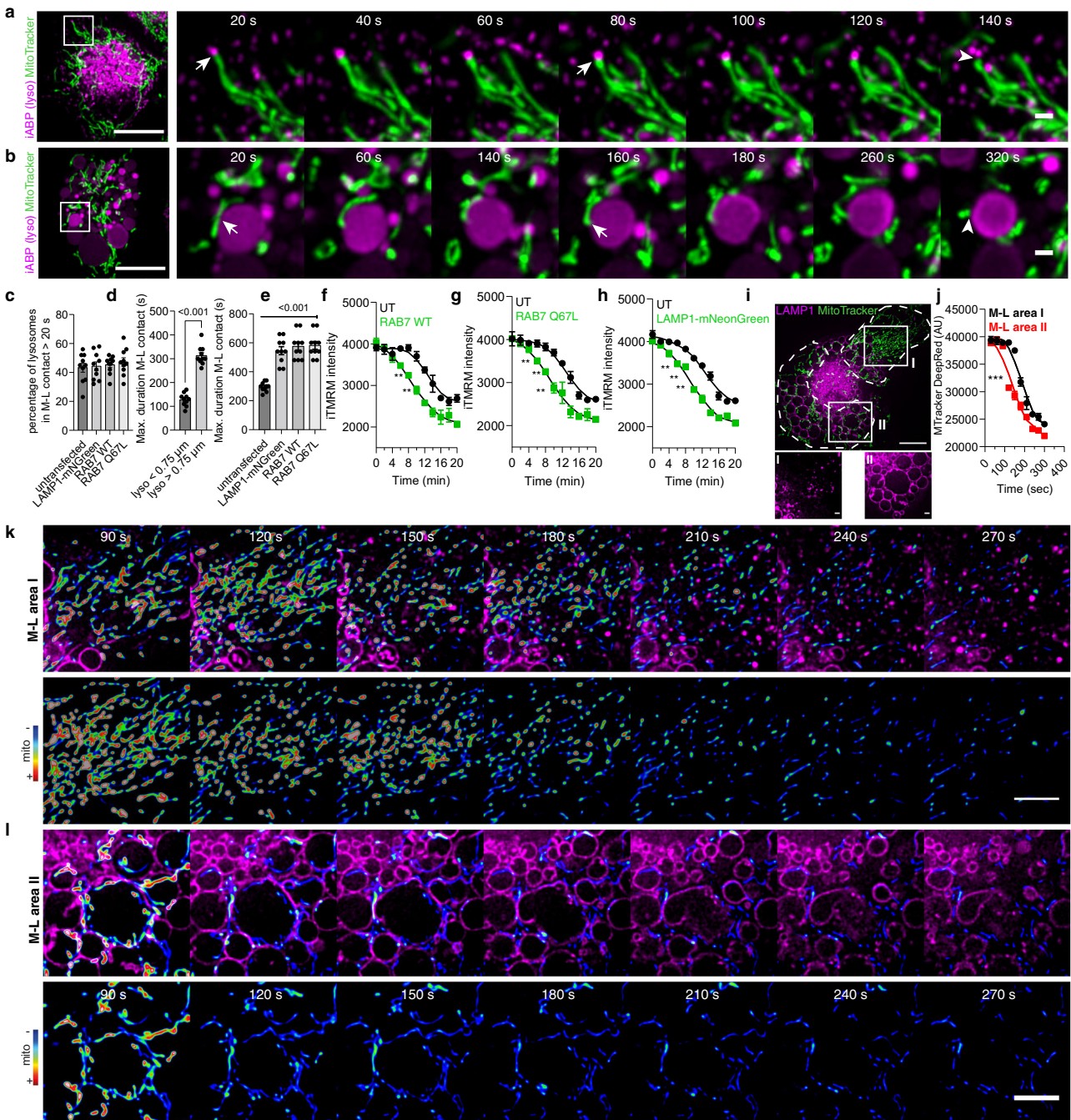

**Fig. 4 | Proteases leakage from damaged lysosomes in the proximity of mitochondria affects mitochondrial activity. a, b** Live-cell super-resolution imaging (20 s time frame) of iPSDM incubated with MitoTracker Green and with the iABP probe showing sequences following a small (**a**) and a large (**b**) lysosome. Scale bars: 10 μm and 1 μm for images and zoom-in, respectively, (see Supplementary Movie 1–2). **c–e** Quantification from the datasets described above and in Supplementary Fig. 4 showing the percentage of lysosomes in contact with a mitochondrion longer than 20 s (**c**), the maximum duration of M-L contact in untransfected iPSDM considering lysosomes smaller than 0.75 μm and larger than 0.75 μm (**d**) or comparing untransfected iPSDM with iPSDM transiently expressing the indicated plasmids (**e**). Data represent the mean ± SEM of at least 10 cells from one out of three independent experiments. **f–h** High content single cell evaluation of iTMRM intensity over time comparing untransfected iPSDM with iPSDM transiently

expressing RAB7 WT GFP (**f**), RAB(Q67L) GFP (**g**) or Lamp1-mNeonGreen (**h**) after the addition of 0.5 mM of LLOMe. Spline curves are shown, a paired *t*-test test was used for comparisons. **i** iPSDM expressing Lamp1-mNeonGreen where the mitochondrial membrane potential evaluation is done in two different M-L areas, where area "I" is characterized by a higher density of small lysosomes in comparison with area "II" as indicated in the bottom panel. Scale bar: 1 μm. **j** Quantification of MitoTracker Deep Red intensity after LLOMe treatment in the indicated M-L areas. **k, l** A time-lapse sequence belonging to M-L area "I" and "II", respectively. The mitochondrial signal is shown as RGB rainbow scale, (see Supplementary Movie 3–5). **p ≤ 0.01; ***p ≤ 0.001. Bar plots represent mean values +/− SEM of at least three independent experiments. Scale bars: 10 μm and 5 μm for images and zoom-in, respectively. Source data are provided as a Source Data file.

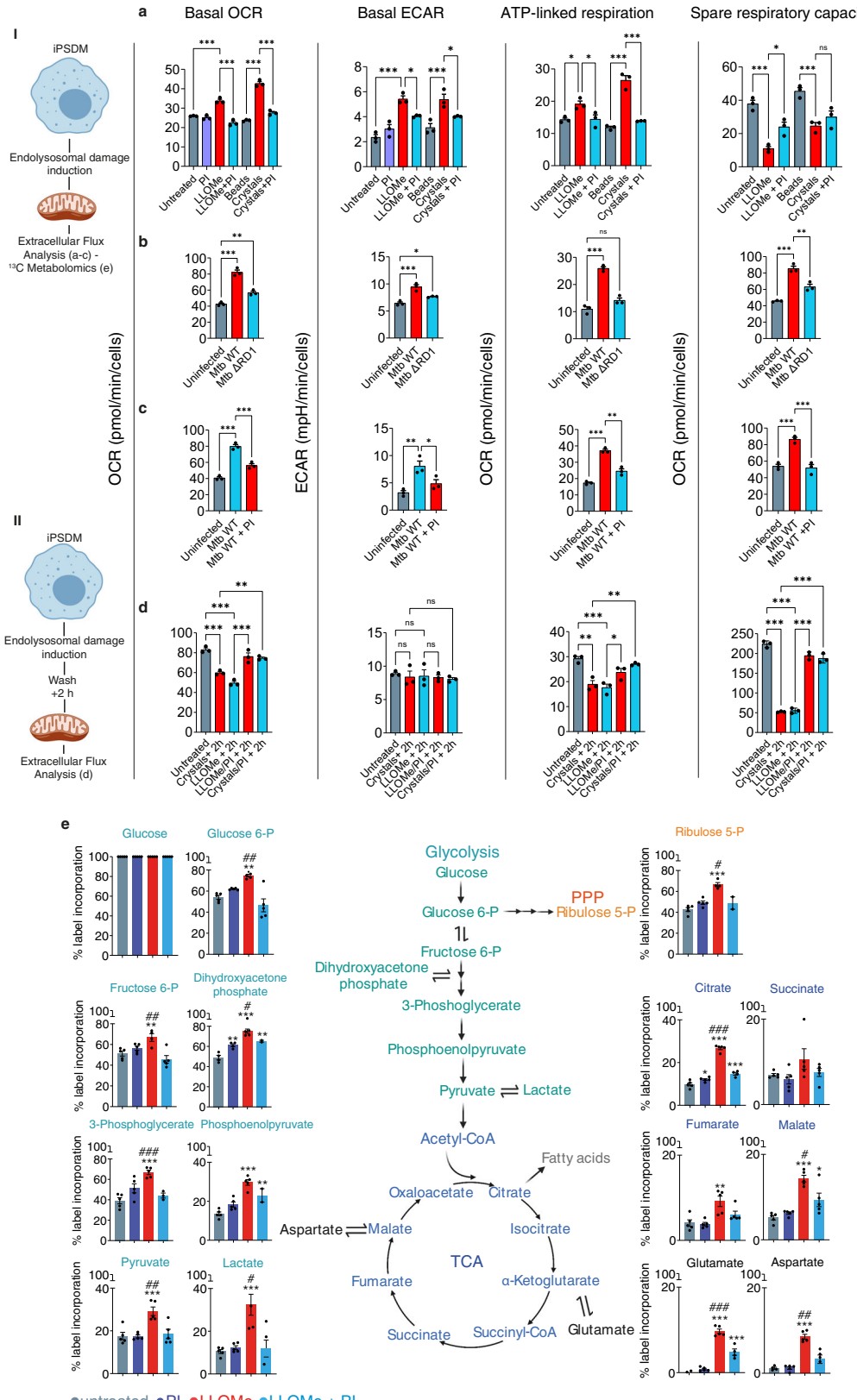

**Nature Communications** | (2022)13:7338

## Lysosomal leakage regulates metabolic and immune responses in lung macrophage populations in vivo

To identify functional outcomes of the immune response to silica crystals of macrophages in the lungs, we conducted single-cell-RNA-seq analysis of BAL samples to identify immune phenotypes and metabolic profiles specifically associated with endomembrane damage in lung macrophage populations. By single-cell transcriptional profiling, we identified seven distinct macrophage populations in BAL-derived samples (M1-M7, Fig. 7a, b, Supplementary Fig. 8a, Supplementary Table 2 and Supplementary Data 3 and 4). A Gene Set Enrichment Analysis (GSEA) of Reactome pathway expression comparing mice treated with silica beads and crystals identified a

**Fig. 5 | Lysosomal protease leakage reprograms macrophage metabolism.**
**a** Extracellular flux analysis (EFA) using Cell Mito Stress Test kit with iPSDM left untreated or treated with LLOMe (0.5 mM, 1 h) or silica crystals (100 µg/mL, 3 h) in presence or absence of PI. **b** EFA of iPSDM uninfected or infected with Mtb ΔRD1 or Mtb WT for 48 h. **c** EFA of iPSDM uninfected or infected with Mtb WT for 48 h in presence or absence of PI. **d** iPSDM treated as in (**a**) but EFA started after 2 h of removing LLOMe or silica crystals treatment. Data show the mean ± SEM of two out of three independent experiments. Values were normalised to cell number.
$^*p ≤ 0.05, ^{**}p ≤ 0.01, ^{***}p ≤ 0.001$, one-way ANOVA with Tukey's multiple

comparisons test. **e** $^{13}$C enrichment of metabolites extracted from iPSDM incubated with [U-13C]glucose and left untreated or treated with 0.5 mM of LLOMe for 1 h in the presence or absence of PI ($n = 5$ technical replicates). "I" and "II" illustrate the time point when EFA started or $^{13}$C enrichment was evaluated. $^*p ≤ 0.05, ^{**}p ≤ 0.01, ^{***}p ≤ 0.0001$, one-way ANOVA with Dunnett's multiple comparisons test vs untreated. $^\#p ≤ 0.05, ^{\#\#}p ≤ 0.01, ^{\#\#\#}p ≤ 0.001$, unpaired $t$-test (LLOMe vs LLOMe + PI). Source data are provided as a Source Data file. See also Supplementary Fig. 5 and Supplementary Data 2.

predominant upregulation of intracellular trafficking pathways, including vesicle-mediated transport and antigen presentation. (Fig. 7c). The cluster M5, characterised by the expression of the alveolar macrophage markers Krt79 and Car4[41], downregulated mitochondrial respiration-related pathways and showed an increase in lipid metabolism gene expression (Fig. 7b, d). Notably, the other two main macrophage populations (clusters M3 and M7) displayed a similar transcriptional profile (Fig. 7d). Conversely, a macrophage subset expressing Cd83, Cd86 and Cxcl16 showed increased levels of OXPHOS-related genes, indicating that not all the BAL macrophage subsets induced a similar mitochondrial transcriptional profile in response to lysosomal damage (Fig. 7b, d). Unlike the main macrophage population (clusters M3, M5 and M7) which are enriched in canonical "M2" polarization markers, such as Chil3 and Mrc1 (Fig.7b and Supplementary Fig. 8a and 9a–c); the macrophage subsets characterized by the expression of pro-inflammatory or canonical "M1" polarization markers such as Ccl4, Cd80 and Cd86 (clusters M1 and M2, Fig. 7b and Supplementary Fig. 8a and 9d–e) did not show reduced TCA cycle and respiratory electron transport pathway activity after silica crystals challenge (Fig. 7d). Intriguingly, and in contrast with the clusters M1 and M2, the macrophage subsets enriched in "M2" polarization markers (particularly, clusters M3 and M5) displayed the highest transcript levels of lysosomal cathepsins (Supplementary Fig. 9f–l), which suggest a correlation between macrophage polarization, lysosomal content and mitochondrial function which supports our in vitro results in iPSDM. In addition, all the silica-crystals macrophage subsets showed increased transcript levels of ROS and RNS pathways and most of the subsets increased transcript levels of the glucose metabolism pathway. Collectively, these results are in accordance with the ROS-driven increase in OCR values and elevated ECAR observed by extracellular flux analysis (Figs. 6h and 7d). These macrophages subsets with altered metabolic pathways were characterized by differing degrees of pro-and anti-inflammatory responses with a consistent upregulation of IL-10 and interferon signalling pathways (Fig. 7e). There was no significant regulation of cell death pathways in the main macrophage populations, consistent with the previous results that do not show compromised cell viability (Fig. 6b and Supplementary Fig. 8b).

To define if macrophage metabolism reprograming after endomembrane damage affected macrophage effector function, we analysed the effect of metabolic inhibitors in our endolysosome membrane damage Mtb infection model. While 2-deoxy-d-glucose (2-DG) inhibits hexokinase activity and consequently, glycolysis, oxamate is a competitive inhibitor of the enzyme lactate dehydrogenase and directs glucose metabolism through OXPHOS[42,43]. Modulation of host cell metabolism had a striking effect in the control of Mtb WT but not in response to Mtb ΔRD1 (that is unable to induce endomembrane damage) (Fig. 7f–h). We observed that 2-DG induced an increase in Mtb WT replication while oxamate restricted Mtb replication in macrophages, in agreement with previous observations[44,45]. However, we did not observe any effect in the host control of Mtb ΔRD1 (Fig. 7f–h), arguing that endolysosomal damage-induced metabolic changes have functional consequences in antimycobacterial host cell responses. Altogether, these results indicate that lysosomal leakage impacts mitochondrial function and reprograms macrophage metabolism with consequences for inflammation and innate immune responses (Fig. 7i).

## Discussion
Given that the process of lysosomal damage and repair is critical for the homeostasis of eukaryotic cells, there is growing interest in trying to understand the fundamental mechanisms that govern how cells metabolically adapt to endomembrane injury. Here, we uncovered a lysosome-mitochondria communication pathway whereby degradation of mitochondrial proteins by lysosomal proteases after endomembrane damage modulates macrophage immunometabolism in homeostatic conditions (Fig. 7i). We postulate this pathway orchestrates a rapid response to endomembrane damage that will metabolically reprogram macrophages to cope with, and eventually resolve, the damage before macrophage viability is compromised. Our results argue this mechanism is precedent to cell death and specific to macrophages. Given that it operates in the absence of ATG7, PRKN or proteasome activity, this mechanism is clearly different from other previously reported endolysosome-dependent mitochondrial degradation pathways[14–17]. In contrast to the targeting of entire dysfunctional mitochondria to mitophagy, this proteolytic degradation of mitochondrial proteins is a relatively rapid process and a direct consequence of lysosomal damage. Although the extent of mitochondrial protein turnover mediated by MDVs in human macrophages is unknown, it has been shown that PRKN negatively regulates MDVs in mouse macrophages[46]. We show here that mitochondrial protein degradation after lysosomal leakage occurs in both PRKN-expressing and PRKN KO macrophages and we did not find evidence of MDVs induction after triggering endomembrane damage with LLOMe, silica crystals or *M. tuberculosis* infection. Proteases residing within mitochondria (mitoproteases) are emerging as pivotal regulators of mitochondrial proteostasis and an increasing number of regulatory functions of mitoproteases have recently been reported[30,47]. We present several lines of evidence showing that mitochondrial protein degradation and functional changes are driven by limited lysosomal leakage and consequent proteolytic activity but not by mitoproteases. Whether secondary metabolic responses mediated by mitoproteases have an effect on the macrophage phenotype after lysosomal damage remains to be investigated.

In agreement with recent evidence showing organelle degradation by cytosolic proteolytic activity in the absence of autophagy[48], it is possible that the lysosomal protease-dependent mechanism we report here can functionally modify other organelles, in addition to the mitochondria, and further studies are needed. Our spatiotemporal live super-resolution imaging analysis highlights that proximity between a lysosome undergoing damage and the mitochondria is required for the changes in mitochondrial activity and function. Consistently, increasing the duration of mitochondria-lysosome contacts during endomembrane damage have a strong impact in the mitochondrial network activity. How this crosstalk between mitochondria and lysosomes affects mitochondrial function in other cell types remains to be elucidated. Of note, our data clearly show that this mitochondria-lysosome interaction only occurs in cells with high levels of lysosomal enzymes such as macrophages. Thus, identifying the cell types with high levels of lysosomal proteases will be important to define the functional consequences of this damaged lysosome-mitochondria crosstalk. Collectively, our metabolic analysis and mitochondrial function studies after triggering lysosomal damage show elevated

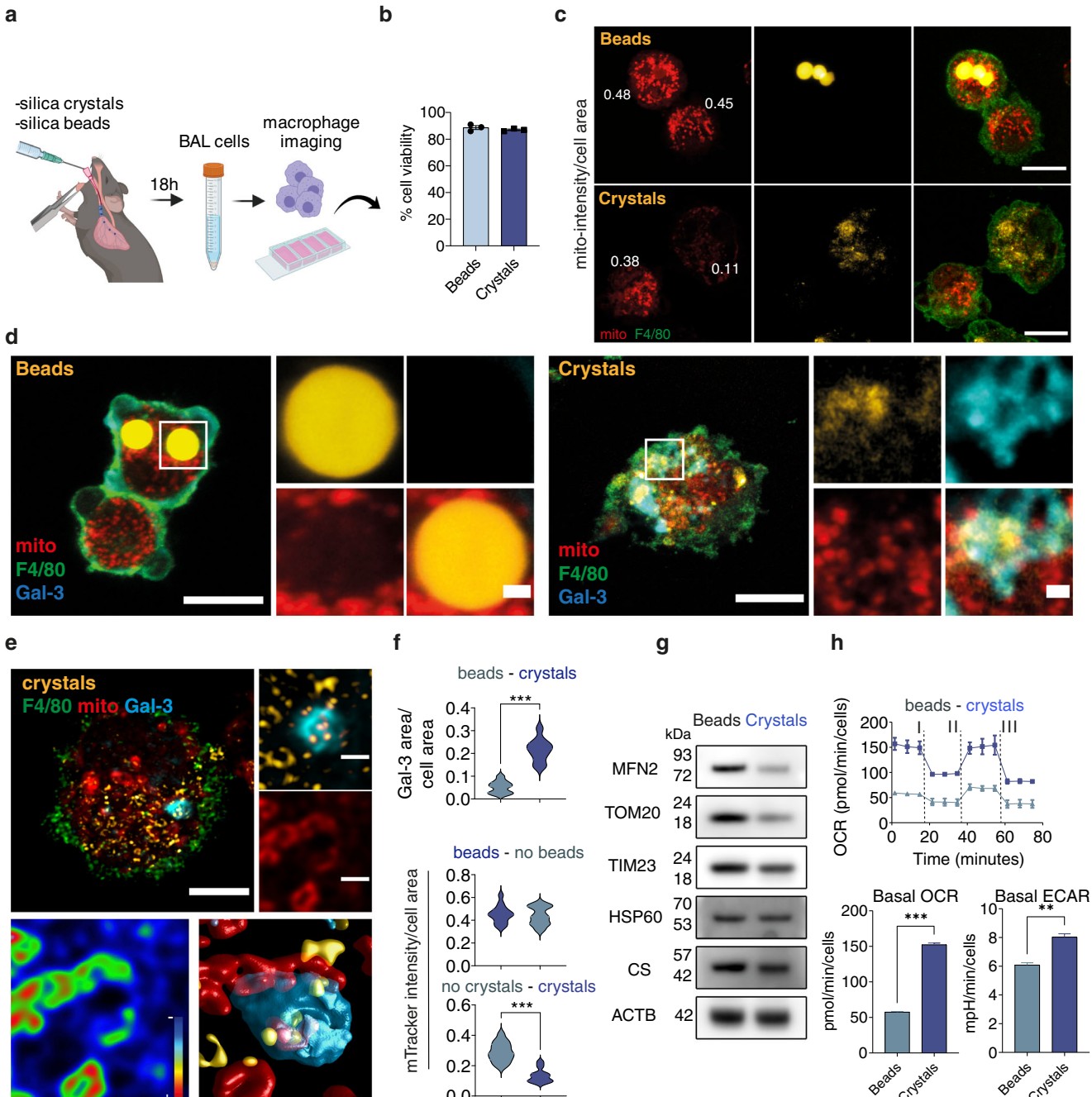

**Fig. 6 | Lysosomal leakage affect mitochondrial activity and macrophage metabolism in vivo. a** Mice were intratracheally intubated with silica crystals or beads and after 18 h BAL fluids obtained. **b** Bar graph shows cellular viability measured by trypan blue exclusion test ($n = 3$ independent experiments). **c** BAL cells were labelled with anti-F4/80 antibody and incubated with MitoTracker Deep Red for live-cell confocal imaging and mitochondrial fluorescence intensity evaluation. $n = 3$ independent experiments. **d** BAL cells were labelled with MitoTracker Deep Red, fixed and stained for F4/80 and Galectin-3 (Gal-3). Crystals were imaged by reflection microscopy. $n = 3$ independent experiments. **e** 3D Confocal imaging analysis of an F4/80+ macrophage with low membrane potential mitochondria in the proximity of a crystals-induced GAL-3-positive endolysosome. **f** Quantitative analysis of MitoTracker intensity and GAL-3 area in F4/80+ macrophages from the BALs. At least 20 cells were counted per condition. **g** Western blot analysis of mitochondrial proteins in BAL cells. **h** EFA shows the OCR values after the addition of Oligomycin (I), FCCP (II) and a mix of rotenone/antimycin (III). The bar graphs show the basal OCR and ECAR values. Data are from one representative experiment out of two. Values were normalised to cell number. An unpaired two-tail t-test test was used for comparisons. **$p \leq 0.01$, ***$p \leq 0.001$. Images shown are z-stack projections. Scale bars, 10 µm and 1 µm for images and zoom-in, respectively. Bar plots represent mean values +/− SEM of at least two independent experiments. Unprocessed blots and Source data are provided as a Source Data file.

production of mitochondrial ROS and impaired OXPHOS activity with macrophages displaying increased glycolytic metabolism. The lipidomics analysis revealed that macrophages also rewired lipid metabolism after endomembrane damage and these results were consistent with our in vivo scRNA-seq evaluation were the macrophage subsets

from the silica crystals-treated mice increased the transcript levels of genes associated with lipid metabolism.

Our data suggest that lysosome luminal contents, including resident proteases, leaked into the mitochondria after lysosome damage. Moreover, our results indicate that limited lysosomal damage in

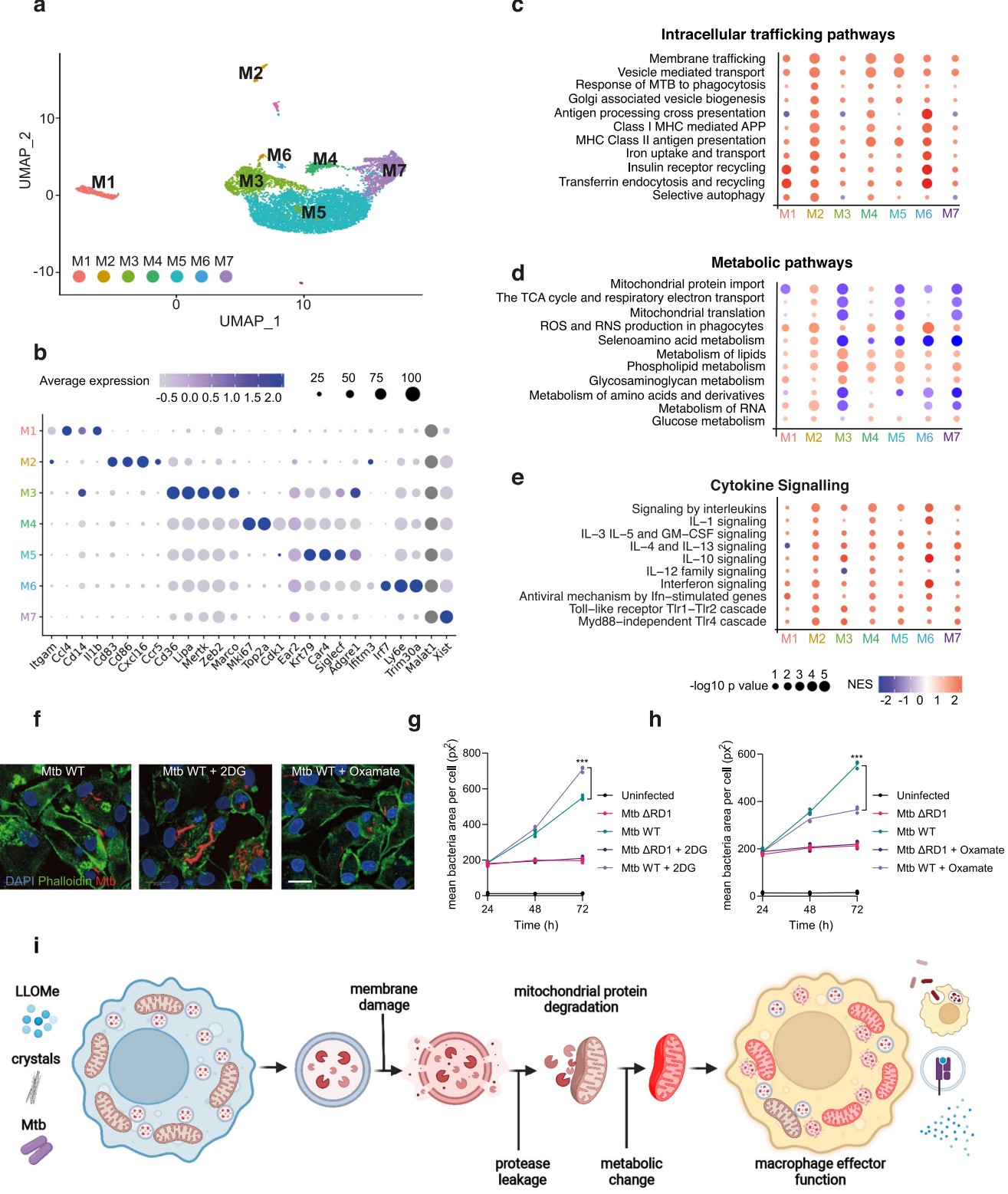

**Fig. 7 | Lysosomal leakage regulates metabolic and immune responses in specific subsets of macrophages in vivo. a** UMAP plot of BAL cells (n = 8723) showing seven identified macrophage clusters (M1-M7). **b** Dot-plot showing expression levels of representative genes for each macrophage cluster. **c**–**e** Intracellular trafficking (**c**), metabolic (**d**) and cytokine signalling (**e**) pathways significantly enriched by the treatment among the different macrophage populations (results show fold change vs beads). **f** Representative images of iPSDM infected with Mtb WT for 72 h in the presence of absence of 2-DG or oxamate. *n* = 3 independent experiments. **g**, **h** iPSDM were infected with Mtb ΔRD1 or Mtb WT and incubated in the presence or absence of 2-DG (**g**) or oxamate (**h**). Graphs show mean bacteria area per macrophage at 24, 48 and 72 h after infection. Data represent the mean ± SEM of three independent biological replicates. One-way ANOVA and Tukey post-test was used for multiple comparisons, \*\*\**p* ≤ 0.001. **i** Scheme summarising the main events leading to macrophage metabolic reprogramming after lysosomal protease leakage. Source data are provided as a Source Data file. Scale bar, 10 µm. See also Supplementary Figs. 8–9, Supplementary Table 2 and Supplementary Data 3 and 4.

macrophages induced the oligomerisation of VDAC1, and inhibition of VDAC1 oligomerisation is required for changes in mitochondrial activity after endomembrane damage. These results agree with recent evidence shown that during oxidative stress, viable cells could form macropores by oligomerization of voltage-dependent anion channels (VDACs) facilitating mtDNA release into the cytosol[27]. While we cannot rule out the contribution of other mitochondrial pores, we envision a scenario where permeabilization of the OMM upon protease leakage, possibly by formation of pores consisting of VDAC oligomers, facilitate the access of cathepsins into the intermembrane space and degrade outer and inner membrane proteins, thereby also compromising electron transport chain function.

Our observations in an in vivo model of silicosis indicate that metabolic rewiring during endomembrane damage is important for the immune response of macrophages. These results agree with recent reports showing that phagosomal rupture and escape of phagosomal contents into the cytosol favours cross-presentation of dead-cell-associated antigens[8,9]. The consequences of protease leakage and modulation of cell metabolism in the context of antigen presentation by myelocytic cells remain to be investigated.

Based on our sc-RNA-seq analysis, the metabolic response to endomembrane damage is differentially regulated in macrophage subsets present in the BALs. Mtb is known to induce metabolic reprogramming of human macrophages[44,45] and we show here this is, in part, due to the ability of this pathogen to inflict membrane damage and protease leakage that alters mitochondrial function. In the context of infection, this transient damaged lysosome-to-mitochondria interactions that are precedent to cell death are likely to be critical for innate immune responses to pathogens. Because the degradation of mitochondrial proteins seems to be restricted to macrophages, our results could explain the different ability of cell types to respond to bacterial infection.

It is becoming clear that endomembrane damage is constantly occurring not only after external stimuli and infection but also as a consequence of membrane lipid oxidation[3,49,50] or protein aggregation[51–53]. In most of the cases, cells efficiently detect and activate pathways to repair the endomembrane damage to survive[1,2]. However, during these damage/repair cycles, proteases and other lysosomal luminal components will leak into the cytosol. Our data suggest that these transient events of lysosomal leakage have a pivotal role regulating mitochondrial function and metabolism to maintain homeostasis. Whereas it is unclear if endomembrane repair mechanisms are affected by aging, we speculate that increasing failures to repair limited endomembrane damage contributes to the phenomenon of mitochondrial decline in the absence of cell death. Our data provide evidence linking lysosomal damage, mitochondrial function and immunometabolism with important consequences for the macrophage immune response.

## Methods

The authors confirm that mice were bred and housed in specific pathogen-free facilities at The Francis Crick Institute. All protocols for breeding and experiments were approved by the Home Office (U.K.) under project license P4D8F6075 and performed in accordance with the Animal Scientific Procedures Act, 1986.

## Plasmids

All DNA constructs were produced using Escherichia coli DH5a (Thermo Fisher Scientific) and extracted using a plasmid midiprep kit from Qiagen. The plasmids used in this study were: RFP-Galectin-3[21], pMitoTimer (Addgene, 52659), pHyPer-dMito (Evrogen, FP942), CystatinB C-GFPSpark (HG10435-ACG, Sino Biological), Lamp1-mNeonGreen (Addgene, 98882), RAB7-GFP[54] and RAB7(QG7L)-GFP[54]. The mCherry-EGFP-NIPSNAP mitophagy reporter was kindly provided by Dr. Anne Simonsen (University of Oslo, Norway).

## Cells

**iPSC and iPSDM culture.** EIKA2 and KOLF2 human iPSCs were sourced from Public Health England Culture Collections (catalogue number 77650059 and 77650100, respectively) and maintained in Vitronectin XF (StemCell Technologies) coated plates with E8 medium (ThermoFisher Scientific). Cells were authenticated by STR profiling upon receipt and are checked monthly for Mycoplasma contamination by PCR. Cells were passaged 1:6 once at 70% confluency using Versene (Gibco). Monocyte factories were set up following a previously reported protocol (van Wilgenburg et al., 2013). Briefly, a single cell suspension of iPSCs was produced with TryplE (Gibco) at 37 °C for 5 min and resuspended in E8 plus 10 μM Y-27632 (Stem Cell Technologies) and seeded into AggreWell 800 plates (StemCell Technologies) with $4 \times 10^6$ cells/well and centrifuged at $100 \times g$ for 3 min. The forming embryonic bodies (EBs) were fed daily with two 50% medium changes with E8 supplemented with 50 ng/ml hBMP4 (Peprotech), 50 ng/ml hVEGF (Peprotech) and 20 ng/ml hSCF (Peprotech) for 3 days. On day 4, the EBs were harvested by flushing out of the well with gentle pipetting and filtered through an inverted 40 μm cell strainer. EBs were seeded at 100–150 EBs per T175 or 250–300 per T225 flask in factory medium consisting of X-Vivo 15 (Lonza) supplemented with Glutamax (Gibco), 50 μM β-mercaptoethanol (Gibco), 100 ng/ml hM-CSF (Peprotech) and 25 ng/ml hIL-3 (Peprotech). These monocyte factories were fed weekly with factory medium for 5 weeks until plentiful monocytes were observed in the supernatant. Up to 50% of the supernatant was harvested weekly and factories fed with 10–20 ml factory medium. The supernatant was centrifuged at $300 \times g$ for 5 min and cells resuspended in X-Vivo 15 supplemented with Glutamax, 50 μM β-mercaptoethanol and 100 ng/ml hM-CSF and plated at $4 \times 10^6$ cells per 10 cm petri dish to differentiate over 7 days. On day 4, a 50% medium change was performed. To detach cells, iPSDM plates were washed once with PBS then incubated with Versene for 15 min at 37 °C and 5% $CO_2$ before diluting 1:3 with PBS and gently scraping. Macrophages were centrifuged at 300 g and plated for experiments in X-VIVO 15.

**Generation of stable MITO-Tag iPSC.** EIKA2-3XMYC-EGFP-OMP25 and EIKA2-3XHA-EGFP-OMP25 iPSCs were generated by ALSTEM, INC (Richmond, CA) by subcloning 3XHA-EGFP-OMP25 and 3XMYC-EGFP-OMP25 gene into the lentiviral expression vector pLenti-EF1a-MCS-PGK-puro. The expression of transgenes was checked by puromycin integration.

Culturing of human iPS cells: Cells were cultured in complete mTeSR1 media with the plate pre-coated with Matrigel at 37 °C 5% $CO_2$. Cells were maintained by splitting every 3 days at a ratio of 1:5 to 1:10. Lentivirus packaging: Briefly, $1 \times 10^7$ HEK293FT cells (Invitrogen R7007) were transfected with lentiviral expression vector and packaged mix next day; the viral supernatant was collected two-day post transfection; after concentration, small aliquots were stored at −80 °C. The titer was measured by qPCR method. The virus titer (ifu/ml) defined by Clontech's Lenti-X qRT-PCR Titration Kit (Cat. # 631235) was:

$$3XHA - EGFP - OMP25 \, \text{lentivirus} \quad 2.72 \pm 0.37 \times 108$$
$$3XMYC - EGFP - OMP25 \, \text{lentivirus} \quad 4.14 \pm 0.35 \times 108$$

Stable cell line generation: ~$1 \times 10^5$ EIKA2 cells were plated per well in a 12-well plate, and the lentivirus at MOI of 2–5 added next day. Cells were treated with puromycin at 1 μg/ml on day 4 and after 5 days the concentration was reduced to 0.25 μg/ml for cell expansion. The cells then were passaged to one 10 cm dish for each line. The cells were frozen at $5 \times 10^5$ cells per vial. These cells are also negative to bacterial and mycoplasma contamination

Transgene expression analysis:

| Primer sequence | |
| --- | --- |
| puro-F | 5'-TGACCGAGTACAAGCCCAC-3' |
| puro-R | 5'-ACACCTTGCCGATGTCGAG-3' |

**Amplicon size:** 192 bp

**Generation of ATG7 and PARKN knockout iPSC.** CRIPSR/Cas9 technology was used to generate ATG7, PARKIN and the double ATG7/PARKN KO iPSC cell lines. The KO strategy was based on using 4 sgRNAs flanking specific gene exons to obtain a deletion of a genomic sequence. The sgRNAs targeting ATG7 and PARKN gene were designed and selected sgRNA considering the lowest off-target score by using WGE CRISPR design tool[l] (www.sanger.ac.uk/htgt/wge/).

Nucleofection of EIKA2 iPSC was performed by using the Amaxa 4D-Nucleofector (V4XP-3024, Lonza, Germany). For each nucleofection, 4 μg of each synthetic chemically modified sgRNAs (Synthego) were used with 20 μg Cas9 Nuclease V3 (IDT). After nucleofection, single clones were manually picked and screened by PCR-based assay. For the double ATG7/PARKN KO, the D4 clone KO for ATG7 has been chosen to proceed with a second nucleofection with the sgRNA and Cas9 for the PARKN gene. All the KO clones were confirmed and selected by PCR.

| sgRNA | | |
| --- | --- | --- |
| **Name** | **ID (WGE)** | **Sequence 5->3** |
| CRISPR1-ATG7 | 950233638 | TTATTTTGAGGTAAACAGTT |
| CRISPR2-ATG7 | 950233647 | TAACATATCCATGCAAAACC |
| CRISPR3-ATG7 | 950233717 | CTCCCTCATAGGTGGACCAC |
| CRISPR4-ATG7 | 950233744 | ATCAGAGTCAATTAGGTGCC |
| CRISPR1_PRKN | 1018304667 | AGGTTCAGTAACATTGTCCC |
| CRISPR2_PRKN | 1018304675 | CGCACAGTTCTGCAGCTGAT |
| CRISPR3_PRKN | 1018304758 | TGTCAGAATCGACCTCCACT |
| CRISPR4_PRKN | 1018304745 | CTGCGAAAATCACACGCAAC |

| PRIMERS | |
| --- | --- |
| **Name** | **Sequence 5->3** |
| GF1_ATG7 | TGTCTGTGTCGAAGAAGGAATAACA |
| GR1_ATG7 | TATAATGCCTTGCACAGTGGCTTG |
| GF1_PRKN | GCTGGTTAAGAAAAGGGGAACATAC |
| GR1_PRKN | CGTGGGTAACTAACTCTGTTTTTCC |

**iPSDM electroporation.** Plasmid DNA was electroporated into iPSDMs using the Neon system (Invitrogen). iPSDM were resuspended at $1.5 \times 10^6$ cells in 100 μl buffer R. 10 μl of cell/1 μg plasmid DNA mix was aspirated into a Neon pipette and electroporated in electroporation buffer 'E' at 1500 V for 30 ms with 1 pulse. Cells were then plated in ViewPlate glass bottom 96-well plates (6005430, PerkinElmer) for high-content analysis or in IBIDI μ-Slide 18-well glass bottom coverslips (Cat# 81817) for confocal imaging studies.

**Human monocyte-derived primary macrophage isolation.** White blood cells were isolated from leukocyte cones (NC24) supplied by the NHS blood and transplant service by centrifugation on Ficoll-Paque Premium (GE Healthcare 17-5442-03) for 60 min at $300 \times g$. Mononuclear cells were washed twice with MACS rinsing solution (Miltenyi 130-091-222) to remove platelets, then remaining red blood cells were lysed by incubation at room temperature with 10 mL RBC lysing buffer

(Sigma R7757) per pellet for 10 min. Cells were washed with rinsing buffer and pelleted once more, then re-suspended in 80 μL MACS rinsing solution supplemented with 1% BSA (MACS/BSA) and 20 μL anti-CD14 magnetic beads (Miltenyi 130-050-201) per $10^8$ cells and incubated on ice for 20 min. Cells were then washed in MACS/BSA by centrifugation, re-suspended in 500 μL MACS/BSA per $10^8$ cells and passed through an LS column (Miltenyi 130-042-401) in the field of a QuadroMACS separator magnet (Miltenyi 130-090-976). The column was washed three times with MACS/BSA, then positively selected cells were eluted, centrifuged, and re-suspended in RPMI 1640 with GlutaMAX and HEPES (Gibco 72400-02), 10% heat-inactivated fetal bovine serum and 50 ng/ml hM-CSF (Preprotech) to a concentration of $10^6$ cells/mL in untreated petri dishes. These were placed in a humidified 37 °C incubator with 5% $CO_2$, with an equal volume of fresh media including hM-CSF added after 3 days. Six days after the initial isolation, differentiated macrophages were detached in 0.5 mM EDTA in ice-cold PBS and $5 \times 10^5$ cells seeded per well of a 12-well plate for Western blot experiments.

**Mtb infection.** Mtb H37Rv WT and Mtb H37Rv ΔRD1 were kindly provided by Prof. Douglas Young (The Francis Crick Institute, UK) and Dr Suzie Hingley Wilson (University of Surrey, UK). Fluorescent Mtb strains were generated as previously reported[55]. E2Crimson Mtb was generated by transformation with pTEC19 (Addgene 30178, deposited by Prof. Lalita Ramakrishnan). Strains were verified by sequencing and tested for PDIM positivity by thin-layer chromatography of lipid extracts from Mtb cultures. Mtb strains were cultured in Middlebrook 7H9 supplemented with 0.2% glycerol, 0.05% Tween-80 and 10% albumin dextrose catalase (ADC). For macrophage infections, Mtb was grown to OD600~0.8 then centrifuged at $2000 \times g$ for 5 min. The pellet was washed twice with PBS, then the pellet was shaken with 2.5–3.5 mm glass beads for 1 min to produce a single-cell suspension. The bacteria were resuspended in 10 ml cell culture medium and centrifuged at $300 \times g$ for 5 min to remove clumps. The OD600 was determined, and bacteria diluted to an appropriate OD for the required multiplicity of infection (MOI) – assuming OD600 = 1 equates to $10^8$ bacteria/ml – before adding to cells in a minimal volume. After 2 h, the inoculum was aspirated, cells washed twice with PBS and fresh culture medium added. Cells were then incubated for appropriate time points before collecting for analysis as described in the sections below. For oxamate and 2-deoxy-D-glucose (2-DG) treatment, fresh culture medium containing sodium oxamate at 0.5 mM (O2751-5G, Sigma) or 2-DG (D8375-1G) at 1 mM was used. For replication studies, iPSDM were infected with a MOI of 1. For all other experiments, cells were infected with a MOI of 2 for 48 h.

**Isolation of mouse bone marrow macrophages.** Primary mouse bone marrow macrophages were isolated as described[56]. The bone marrow of mice from the C57BL/6J (WT), CtsB−/−, CtsL−/− and CtsS−/− background was isolated, and the cells were plated on sterile microbiology uncoated 9 cm petri dish in RPMI supplemented with 10% FCS and 20% L929 fibroblast supernatant. Cells were differentiated for 6 days at 37 °C in 5% $CO_2$ atmosphere with replacement of 70–80 % of media every 48 h. BMM were collected in ice-cold PBS and plated in RPMI + 10% FCS for experiments.

**HeLa and HEK293-T cell lines culture.** HeLa (ECACC 93021013) and HEK 293-T (ECACC 12022001) cells were grown in DMEM high glucose (Gibco) supplemented with 10% heat-inactivated fetal bovine serum, 2 mM glutamine and maintained at 37 °C and 5% $CO_2$. $4 \times 10^5$ cells were seeded per well of a 12-well plate for Western blot experiments.

**LLOMe treatment.** A 333 mM stock of LLOMe (Cat# 4000725, Bachem) was prepared in ethanol and frozen at −20 °C in tightly sealed tubes. For LLOMe treatment, the medium was replaced with X-VIVO15

containing 0.5 mM of LLOMe. The cells were stimulated for 1 h, after which they were processed for downstream applications. 0.3% ethanol in X-VIVO 15 was used in all control samples.

**Silica treatment.** A solution of crystalline silica (MIN-U-SIL-15, US Silica) at 100 µg/mL was prepared in X-VIVO 15 and the cells were stimulated for 3 h, after which they were processed for downstream applications. As a control, 3 µm silicon dioxide beads (66373-5ML-F, Sigma) or 3 µm red-fluorescent silica beads (40-00-303, Micromod) were used.

**Protease and proteasome inhibitor treatment.** The protease inhibitor cocktail (P1860-1ML, Sigma) was used at a 1:400 dilution and the inhibitor CA-074 methyl ester (CA-074 Me) (S7420, SelleckChem) was used at 50 µM. The proteasome inhibitor bortezomib was used at 5 nM. Unless otherwise specified, all the protease inhibitors were incubated simultaneously with the indicated treatments in X-VIVO 15 media.

**Mitoprotease inhibitors treatment:** the inhibitors 1,10-phenanthroline (o-Phe) (P9375-1G, Sigma), TPEN (P4413-50MG, Sigma) and A2-32-01 (CLpPi) (HY-111532, SelleckChem) were resuspended in DMSO and iPSDM pre-treated for 6 h using the following concentrations o-Phe (1 mM, 6 h); TPEN (0.2 mM) and A2-32-01 (CLpPi) (50 µM). After that, iPSDM were left untreated or treated with LLOMe (0.5 mM, 1 h) and samples processed for WB or mitochondrial membrane potential evaluation as described below.

VDAC1 and BAX oligomerization inhibitor treatment: iPSDM were pre-treated using the VDAC1 oligomerization inhibitor VBIT-4 (S3544, SelleckChem) at 10 µM for 6 h or the BAX oligomerisation inhibitor BAI (HY-103269, MedChemExpress) at 2 µM for 6 h.

**Western blotting.** For lysis, cells were washed once with PBS, and lysed on ice in RIPA buffer (Millipore) containing complete, EDTA-free protease inhibitor (Roche). The samples were boiled at 95–100 °C for 5 min in LDS sample buffer and reducing agent (NuPAGE, Life Technologies) and run on a NuPAGE 4–12% Bis-Tris gel (Life Technologies). The gels were transferred onto a PVDF membrane using an iBlot 2 Dry Blotting System (Thermo Fischer), program P0. Membranes were blocked in 5% skimmed milk powder in PBS plus 0.05% Tween20 (PBS-T) for 1 h at room temperature then incubated with primary antibody overnight at 4 °C. Membranes were washed in PBS-T and incubated with HRP-conjugated secondary antibodies for 1 h at room temperature. Membranes were developed with enhanced chemiluminescence reagent (BioRad) and imaged on an Amersham GE Imager. Antibodies used were: anti-HSP60 (12165S), anti-TOM20 (42406S) and anti-ATG7 (8558S) from Cell Signalling Technology, anti-TIM23 (611222) from Becton Dickinson, anti-CTSB (sc-365558), anti-CTSC (sc-74590), anti CTSL (sc-390385), anti-CS (sc-390693), anti-PRKN (sc-32282) from Santa Cruz, anti-MFN2 (ab56889) from Abcam and Santa Cruz (sc-515647), anti-HA (H6908) from Sigma, anti-YME1L (11510-1-AP), anti-CLPP (15698-1-AP), anti-LONP1 (15440-1-AP) and anti-VDAC1 (55259-1-AP) from Proteintech, anti-OMP25 (PA5-51471) from Invitrogen and HRP-conjugated anti-mouse (W4021) and anti-rabbit (W4011) antibodies from Promega. All antibodies were used at 1:1000 dilution with the exception of HRP-conjugated antibodies that were used at 1:10,000 dilution.

**Cross-linking experiments:** $10^6$ iPSDM were harvested after the indicated treatment and treated as previously described[57]. Briefly, cells were incubated with the cross-linking reagent (ethylene glycol bis(succinimidyl succinate)) (EGS, 21565, Thermo Fischer) at 250 µM for 15 min at 30 °C in PBS pH 8.2. After that, samples were subjected to SDS-PAGE and immunoblotting using anti-VDAC1 antibody.

**Liquid chromatography-mass spectrometry, lipidomics.** $10^6$ iPSDM were seeded in each well of a 6-well plate in RPMI medium without glucose (Thermo Fischer 11879020) containing protein and glucose

concentrations similar to XVIVO15 medium (2.7 g/l and 6.7 g/l, respectively). Ultra pure bovine serum albumin (Cell Signalling Technology 9998) was used as a protein source and iPSDM were incubated with media containing 13C-labelled glucose (Cambridge Isotope Laboratories CLM-1396-5) and the described treatments for 1 h before continuing with downstream sample processing. Macrophages were washed with ice-cold MACS rinsing buffer and then incubated in 500 µL rinsing buffer for 10 min. Macrophages were then detached by scraping, and three wells pooled for each technical replicate. Macrophages were pelleted at $300 \times g$ for 5 min in micro-centrifuge tubes, then re-suspended in 900 µL ice-cold chloroform:methanol (2:1, v/v). Cells were vortexed until dispersed, and spun at $13,200 \times g$ for 10 min at 4 °C. The extracts were transferred to new tubes and dried under nitrogen. The cell pellet was re-extracted with 450 µL methanol:water (2:1, v/v) containing 1 nmol of scyllo-inositol (Sigma) as an internal standard, processed as above, the extract added to the first dry extract and dried again. Metabolites and lipids were partitioned using chloroform:methanol:water (1:3:3, v/v/v), and the organic phase was dried and lipids re-suspended in butanol/methanol (1:1, v/v) containing 5 µM ammonium formate. The LC-MS method was adapted from Amiar et al.[58]. Lipids were separated by injecting 10 µL aliquots onto a 2.1 × 100 mm, 1.8 µm C18 Zorbax Elipse plus column (Agilent) using a Dionex UltiMate 3000 LC system (Thermo Scientific). A 20 min elution gradient of 45–100% Solvent B was used, followed by a 5 min wash of 100% Solvent B and 3 min re-equilibration, where Solvent A was 10 mM ammonium formate in water (Optima HPLC grade, Fisher Chemical) and Solvent B was water:acetonitrile:isopropanol, 5:20:75 (v/v/v) with 10 mM ammonium formate (Optima HPLC grade, Fisher Chemical). Other parameters were as follows: flow rate 600 µL/min; column temperature 60 °C; autosampler temperature 10 °C. MS was performed with positive/negative polarity switching using a Q Exactive Orbitrap (Thermo Scientific) with a HESI II probe. MS parameters were as follows: spray voltage 3.5 kV and 2.5 kV for positive and negative modes, respectively; probe temperature 275 °C; sheath and auxiliary gases were 55 and 15 arbitrary units, respectively; full scan range: 150–2000 $m/z$ with settings of auto gain control (AGC) target and resolution as balanced and high ($3 \times 10^6$ and 70,000) respectively. Data was recorded using Xcalibur 3.0.63 software (Thermo Scientific). Mass calibration was performed for both ESI polarities before analysis using the standard Thermo Scientific Calmix solution. To enhance calibration stability, lockmass correction was also applied to each analytical run using ubiquitous low-mass contaminants. To confirm the identification of significant features, pooled quality control samples were run in data-dependent top-N (ddMS2-topN) mode, with acquisition parameters as follows: resolution of 17,500, auto gain control target under $2 \times 10^5$, isolation window of $m/z$ 0.4, and stepped collision energy 10, 20 and 30 in HCD (high energy collisional dissociation) mode. Qualitative and quantitative analyses were performed using Free Style 1.7 (Thermo Scientific), Progenesis (Nonlinear Dynamics) and LipidMatch[59]. Lipid abundance was normalised to total ion signal.

**Gas chromatography-mass spectrometry, metabolomics.** GC-MS analysis was performed following published protocols[60]. Samples were prepared as above, the polar fraction of the biphasic extract dried and washed twice with methanol. For derivatisation, metabolites were incubated with methoximation (Sigma, 20 µl, 20 mg/ml in pyridine) overnight followed by trimethylsilylation (20 µl of N,O-bis(trimethylsilyl)trifluoroacetamide reagent (BSTFA) containing 1% trimethylchlorosilane (TMCS), Thermo Fisher). Samples were analysed in an Agilent 7890B-7000C GC–MS system. Splitless injection (injection temperature 270 °C) onto a DB-5MS (Agilent) was used, using helium as the carrier gas, in electron ionization (EI) mode. The initial oven temperature was 70 °C (2 min), followed by temperature gradients to 295 °C at 12.5 °C per min and then to 320 °C at 25 °C per min (held for 3 min). Scan range was m/z 50-550. Data analysis was performed using

our in house-developed software MANIC (version 1.0), based on the software package GAVIN[61]. Label incorporation was calculated by subtracting the natural abundance of stable isotopes from the observed amounts. Total metabolite abundance was normalised to the internal standard.

## Imaging

### Confocal microscopy imaging

**Galectin-3 puncta staining.** After the indicated treatments, cells were washed once with PBS and fixed with ice-cold methanol for 5 min at −20 °C, followed by one wash with PBS and blocking in 5% BSA in PBS for 20 min at RT.

After one wash with PBS, cells were incubated with Alexa Fluor 488 anti-mouse/human Mac-2 (Galectin-3) antibody (125410, Biolegend) or alternatively with Brilliant Violet 421 anti-mouse/human Galectin-3 antibody (125416, Biolegend) for 30 min at RT. After three more washes with PBS, when required, nuclear staining was performed using 300 nM DAPI (Life Technologies, D3571) in PBS for 5 min. One final wash with PBS was performed before mounting the coverslips on glass slides using DAKO mounting medium (DAKO Cytomation, S3023) or acquiring images in PBS when using Ibidi slides. Antibodies were used at 1:200 dilution. Images were acquired using a Leica SP8 confocal microscope and Galectin-3 puncta and/or Galectin-3 positive area per cell was evaluated using the image analysis software FIJI/ImageJ.

**Mitophagy evaluation.** After the indicated treatments, mCherry-EGFP-NIPSNAP-transfected iPSDM were fixed with 4% methanol-free PFA in PBS for 15 min. After three washes with PBS, cells were imaged in a Leica SP8 confocal microscope and mCherry only puncta were determined as described for Galectin-3 evaluation. As a positive control, IPSDM were treated with 20 μM of Carbonyl cyanide m-chlorophenyl hydrazine CCCP (Sigma, C2759) in X-VIVO 15 for 3 h.

**Mitochondrial membrane potential evaluation of BAL cells.** BAL cells were incubated for 20 min at 37 °C and 5% $CO_2$ with a 100 nM solution MitoTracker Deep Red FM in DMEM supplemented with 10% heat-inactivated FCS. After that, cells were washed with fresh media and imaged lived or after methanol fixation (see Galectin-3 puncta analysis) in a Leica SP8 confocal microscope.

For silica crystals detection, confocal reflection microscopy was combined with fluorescence microscopy on a Leica SP8 confocal laser-scanning microscope. Reflection was captured by placement of the detector channel directly over the wavelength of the selected laser channel for reflection light capture and the microscope was set to allow 5–15% of laser light into the collection channel.

For 3D imaging, spatial deconvolution and 3D surface-rendering reconstruction, z-stack slices were defined each 200 nm and images were processed using Huygens Essential Software (Scientific Volume Imaging B.V., Netherlands).

Mitochondrial-derived vesicles evaluation: After the indicated treatment, iPSDM were fixed with PFA 4% and immunostained against Tom20 (1:100 dilution) (Invitrogen) and PDHE1 (1:100 dilution) (Proteintech). Z-stacks were acquired using a Leica SP8 confocal microscope with a 60× objective and were analysed using FIJI/ImageJ. Vesicles single-positive for TOM20 or PDHE1 were counted after manual single channel inspection. At least 20 cells per condition were analysed in two independent experiments. As a positive control, iPSDM were treated with glucose oxidase (50 mU/ml GO, 1 h) (Sigma) as previously described[17].

**High content live-cell imaging.** 30,000 iPSDM were seeded into a ViewPlate glass bottom 96- well plate and treated with LLOMe, Silica or infected with *Mycobacterium tuberculosis* as described above. The plate was sealed with parafilm and placed in a pre-heated (37 °C) Opera Phenix microscope with 40× or 60× water-immersion lens (Perkin-nElmer) with 5% $CO_2$.

Capture settings were: Image-iT TMRM Reagent (I34361, Thermo Fischer) and MitoTracker Red CMXRos (M7512, Thermo Fischer) were excited with the 561 nm laser at 10% power with 100 ms exposure. MitoTracker Deep Red FM (M22426, Thermo Fischer), iABP probe and Mtb E2crimson were excited with the 640 nm laser at 10% power with 100 ms exposure. The mitoTimer construct was excited with the 488 nm laser and the 561 nm laser at 10% power and 100 ms exposure. The pHyPer-dMito construct was excited with the 405 and 488 nm lasers and emission was collected at 510 nm for both excitations. At least 20 fields per well were imaged in all the experiments. Images were acquired at 1020 ×1020 pixels using Harmony 4.9 high content imaging and analysis software (PerkinElmer). Cystatin B c-GFPSpark-tag, RAB7-GFP, RAB7(Q67L)-GFP and Lamp1-mNeonGreen- expressing cells were excited using the 488 nm laser at 10% power with 50 ms exposure. Hoechst H33342 (H3570, Thermo Fischer) was excited using the 405 nm laser at 15% power with 100 ms exposure.

**Super-resolution live-cell imaging.** GAL3-RFP-, RAB7-GFP-, RAB7(Q67L)-GFP-, or Lamp1-mNeonGreen-transfected iPSDM were incubated with MitoTracker Deep Red FM for 30 min at 37 °C and 5% $CO_2$. After two washes with PBS, cells were left untreated or treated with LLOMe or silica crystals (200 μg/mL) and imaged on a VT-iSIM super resolution imaging system (Visitech International), using an Olympus IX83 microscope, 150x/1.45 Apochromat objective (UAPON150XOTIRF), ASI motorised stage with piezo Z, and 2x Prime BSI Express scientific CMOS cameras (Teledyne Photometrics). For M-L experiments with untransfected iPSDM, cells were pre-incubated with the iABP probe (1 μm, 3 h) and then incubated with MitoTracker Green FM for 30 min at 37 °C and 5% $CO_2$.

Simultaneous GFP and Cy5 imaging was done using 488 nm and 640 nm laser excitation and ET525/50 m and ET690/50 m emission filters (Chroma), respectively. Z-stacks (100 nm z-step) were acquired every 10 seconds. The microscope was controlled with Micro-Manager v2.0 gamma software[62]. Image processing and deconvolution was done using Huygens Essential software (Scientific Volume Imaging B.V, Netherlands).

### Electron microscopy imaging

**Sample preparation for transmission electron microscopy (TEM) analysis.** After the indicated treatments, cells were washed once with 200 mM HEPES pH 7.4 and fixed by adding double-strength fixative (2.5% Glutaraldehyde and 8% Formaldehyde in 200 mM HEPES, pH 7.4) to the culture medium for 30 min at room temperature, then replace with 1.25% Glutaraldehyde and 4% Formaldehyde in 200 mM HEPES overnight at 4 °C. Samples were processed in a Biowave Pro (Pelco, USA) with use of microwave energy and vacuum. Briefly, cells were twice washed in HEPES (Sigma-Aldrich H0887) at 250 W for 40 s, post-fixed using a mixture of 2% osmium tetroxide (Taab, O011) 1.5% potassium ferricyanide (Taab, P018) (v/v) at equal ratio for 14 min at 100 W power (with/without vacuum 20"Hg at 2-min intervals). Samples were washed with distilled water twice on the bench and twice again in the Biowave 250 W for 40 s. Samples were then stained with 2% aqueous uranyl acetate (Agar scientific AGR1260A) in distilled water (w/v) for 14 min at 100 W power (with/without vacuum 20" Hg at 2 min intervals) then washed using the same settings as before. Samples were dehydrated using a step-wise ethanol series of 50, 75, 90 and 100%, 250 W for 40 s per step, then lifted from the tissue culture plastic with propylene oxide, washed four times in dry acetone and transferred to 1.5 ml microcentrifuge tubes. Samples were infiltrated with a dilution series of 50, 75, 100% of Ultra Bed Low Viscosity Epoxy (EMS) resin to acetone mix and centrifuged at $600 \times g$ between changes. Finally, samples were cured for a minimum of 48 h at 60 °C before trimming and sectioning.

**Sectioning and imaging.** Ultrathin sections (~60 nm) were cut with an EM UC7 Ultramicrotome (Leica Microsystems) using an oscillating ultrasonic 35o diamond Knife (DiaTOME) at a cutting speed of 0.6 mm/s, a frequency set by automatic mode and a voltage of 6.0 volts and placed on ultrathin formvar/carbon coated 100 mesh grids (EMS, FCF100-Cu-UA). Images were acquired using a 120 kv Tecnai G2 Spirit BioTwin (FEI company) with OneView Camera (Gatan).

**Image analysis**
**Galectin-3 puncta analysis.** A threshold was applied using the sequence Image › Adjust › Threshold and then puncta or area in the segmented image was determined using the menu command Analyze › Analyze particles. Size was restricted to particles >0.1 μm and the circularity restricted to values between 0.4 and 1.

**Mitochondrial fluorescence analysis.** Mitochondrial fluorescence intensity was quantified at the single mitochondrial level and analysed as the mean mitochondrial fluorescence intensity per cell. Single-mitochondrial and single-cell segmentation were done in Harmony 4.9 software using SER texture building block and nuclear staining, respectively. Mtb-infected iPSDM were identified after segmentation of bacteria area per cell, as indicated below (Mtb replication analysis) and shown in Supplementary Fig. 3a, b. For GAL-3-proximity analysis, areas with 1 μm diameter were created around GAL-3-positive puncta using the find spot and resize region module. These regions were then segmented for mitochondrial analysis as previously described. CystatinB-GFP Spark, RAB7-GFP, RAB7(Q67L)-GFP and Lamp1-mNeonGreen- expressing IPSDM were single-cell segmented based on the mean green (500–550 nm emission) intensity per cell. One cell was considered positive when the mean green fluorescence intensity was greater than four times the background value. The cells that resulted negative were considered untransfected. For the mitoTimer and pHyPer-dMito evaluation, the red/green (561/488) and the $GFP_{405\ exc}$/$GFP_{488\ exc}$ intensity mitochondrial ratios were determined and the mean per cell quantified. pHyPer-dMito transfected cells were single-cell segmented using a gaussian filter and cell mask building block based on mitochondrial staininig due to incompatibility of the pHyPer-dMito construct with blue fluorescent nuclear dyes.

**Mitochondrial membrane potential analysis.** After the indicated treatments, cells were washed once with PBS and incubated with a 1:1000 solution of Image-iT TMRM Reagent, or a 100 nM solution of MitoTracker Deep Red FM or MitoTracker Red CMXRos for 20 min at 37 °C and 5% $CO_2$. For all the experiments, solutions were made using X-VIVO 15. Nuclear staining was done using 300 ng/mL of Hoechst H33342 and incubated simultaneously with the mitochondrial probes. After that, cells were gently washed with PBS and replaced with fresh media before imaging acquisition. Fluorescence intensities were quantified after single-mitochondrial and single-cell segmentation as described before.

For live-cell super resolution analysis, quantification was done following GAL-3+ vesicles in contact with mitochondria for at least 30 s and measuring MitoTracker Deep Red intensity in the mitochondrion area adjacent or in contact with GAL-3 and in the mitochondrion areas where no contacts or GAL-3 signal was observed. The intensity values were obtained from polygonal ROIs of 0.2–0.4 μm² using FIJI. At least 5 cells were imaged per condition and experiment.

**Imaging of cathepsin activity.** iPSDM, iPSC, HEK293T or HeLa cells were incubated with 1 μM solution of the iABP™ Smart Cathepsin Imaging Probe (40200-100, Vergent Bioscience) for 3 h at 37 °C and 5% $CO_2$. For M-L contact evaluation, iPSDM were then incubated with 100 nM solution of MitoTracker Green FM. High-content imaging and evaluation of lysosomal activity and content: cells were single-cell segmented based on nuclear staining and lysosomes segmented using

the Find Spots and Morphology Properties modules of Harmony 4.9 software. After that lysosomal intensity values (650–760em) and cellular morphology parameters were quantified.

**Live-cell super-resolution analysis of mitochondria (M)–lysosomes (L) contact.** Image analysis was based on a previously described study[63]. Briefly, lysosomes were detected based on iABP staining or the indicated fluorescent constructs. Mitochondria were detected based on MitoTracker Green or Deep Red FM staining. iPSDM were imaged each 20 or 30 s interval for at least 10 min. Individual mitochondrial areas were categorized as in contact with a lysosome (or negative) for showing pixel overlap with the lysosome signal for >20 s. The percentage of lysosomes in contact was quantified as the percentage of vesicles that formed contacts with mitochondria divided by the total number of vesicles in the region of interest. For lysosomal size discrimination, at least nine different events were manually tracked per cell and condition. Mitochondrial and lysosomal segmentation was done using automatic image thresholding (Otsu's method) in FIJI. At least 5 cells were imaged per condition and experiment, with a total of three independent experiments.

**Mtb replication analysis.** Images of Mtb-infected iPSDM were acquired on an Opera Phenix microscope using a 40× objective with at least 20 fields of view per well (with three wells per condition) and analysed in Harmony 4.9. Cells were segmented based on DAPI, excluding any cells touching the edge of the imaged area. Bacteria were detected using the 'Find Spots' building block of Harmony. The total bacterial area in each cell was then determined. Data were exported and analysed in RStudio to calculate the mean Mtb area per cell for each condition at each timepoint, with all three wells pooled. At least three independent experiments were done per condition and timepoint.

**Mitochondrial analysis by TEM.** Using systematic random sampling, a minimum of 19 cells from each condition were identified with a stereological test grid. Micrographs were taken at ×4800 and ×9300 magnification and their mitochondria manually assessed for evidence of vesicle formation. Mitochondria assessed by condition was as follows; uninfected 149, LLOME-treated 121 and Silica crystals-treated 149.

**Stereological analysis of LLOMe-treated iPSDM.** The mitochondria from at least 88 different cells, per condition were imaged at ×3900 magnification by systematic and random sampling with use of a stereological test grid. The subcellular localization of 5 nm BSA Nanogold tracer (EMS), was determined from images taken at minimum magnification of ×10,000. The following criteria were followed for the assessment of subcellular localisation: (a) internalised nanogold; gold particles were found to be either within the matrix of the mitochondrion, or with the two phospholipid bilayers forming the mitochondrial cristae (b) cytosolic nanogold; gold particles were surrounded by ribosomes, representing the cytoplasm with no indication of the mitochondrial membrane.

**Seahorse-based metabolic flux analysis.** For BAL experiments and Mtb-infected iPSDM, cells were seeded onto Seahorse XFp Cell Culture Miniplates (103025-100, Agilent Technologies) and assayed on a Seahorse XFp Analyzer (Agilent Technologies). Oxygen consumption rates (OCR) and extracellular acidification rates (ECAR) were measured in XF DMEM assay medium with pH adjusted to 7.4 (103680-100, Agilent Technologies) containing 10 mM glucose (103577-100, Agilent Technologies), 2 mM L-glutamine (103579-100, Agilent Technologies) and 1 mM sodium pyruvate (103578-100, Agilent Technologies). To investigate mitochondrial respiration and energetic phenotypes a Seahorse XFp Cell Mito Stress Test kit (103010-100, Agilent Technologies) was used. The injection strategy was as follow, first: oligomycin (1 mM at

final concentration), second: carbonyl cyanide 4-(trifluoromethoxy) phenylhydrazone (FCCP) (1 mM at final concentration), and third: rotenone and antimycin A (0.5 mM at final concentration).

For LLOMe and Silica-treated iPSDM, cells were seeded onto XF96 cell culture microplates (101085-004, Agilent Technologies) and assayed on a Seahorse XFe96 Analyzer (Agilent Technologies) on a OCR and ECAR were measured using a Seahorse XF Cell Mito Stress Test kit (103015-100, Agilent Technologies) as described above.

After finishing the assay, cells were fixed with PFA 4% for 15 min. After that, nuclei were stained with DAPI and imaged using an EVOS microscope (Thermo Fischer). The Analyse Particles command from ImageJ was used for nuclei quantification and when required, cell number normalisation was performed.

The WAVE software, version 2.6.1 (Agilent Technologies) was used for further data analysis.

**Nanostring gene expression analysis in iPSDM.** Approximately 150 ng of total RNA were hybridized to a Metabolic Pathways Panel for profiling 780 human genes in a final volume of 15 μl at 65 °C for 22 h according to manufacturer's protocol (NanoString Technologies, Inc., Seattle, WA, USA). Gene expression profiling was measured on the NanoString nCounter™ MAX system. In particular, hybridized samples were processed on the NanoString nCounter™ Preparation Station using the high-sensitivity protocol, in which excess Capture and Reporter Probes were removed and probe-transcript complexes were immobilized on a streptavidin-coated cartridge and data collected on an nCounter digital analyzer (NanoString), following manufacturer's instructions.

Background level was determined by mean counts of eight negative control probes plus two standard deviations. Samples that contain <50% of probes above background, or that have imaging or positive control linearity flags, were excluded from further analysis. Probes that have raw counts below background in all samples were excluded from differential expression analysis to avoid false positive results. Data were normalized by geometric mean of housekeeping genes. All statistical analyses were performed on log2 transformed normalized counts.

Data were analysed by ROSALIND® (https://rosalind.onramp.bio/), with a HyperScale architecture developed by ROSALIND, Inc. (San Diego, CA). Pre-processing and normalization of the raw counts was performed with nSolver Analysis Software v4.0 (www.nanostring.com). The 6 spiked-in RNA Positive Control and the 8 Negative controls present in the panel were used to confirm the quality of the run. Fold changes and pValues are calculated using the optimal method as described in the nCounter® Advanced Analysis 2.0 User Manual. P-value adjustment is performed using the Benjamini-Hochberg method of estimating false discovery rates (FDR). Heatmaps of differentially expressed genes were done using Morpheus (https://software.broadinstitute.org/morpheus/).

**MITO-Tag sample processing and proteomics**
**Sample preparation.** MITO-Tag samples were eluted in 5% sodium dodecyl sulfate (SDS) in 50 mM triethylammonium bicarbonate (TEAB) pH 7.5 and sored at -80 °C until further use. Samples were sonicated using an ultrasonic homogenizer (Hielscher) for 1 minute and further processed for proteomic analysis. Samples were reduced by incubation with 20 mM tris(2-carboxyethyl)phosphine for 15 minutes at 47 °C, and subsequently alkylated with 20 mM iodoacetamide for 30 minutes at room temperature in the dark. Proteomic sample preparation was performed using the suspension trapping (S-Trap) sample preparation method[64,65], with minor modifications as recommended by the supplier (ProtiFi™, Huntington NY). Briefly, 2.5 μl of 12% phosphoric acid was added to each sample, followed by the addition of 165 μl S-Trap binding buffer (90% methanol in 100 mM TEAB pH 7.1). The acidified samples were added, separately, to S-Trap

micro-spin columns and centrifuged at 4000 × g for 1 min until all the solution has passed through the filter. Each S-Trap micro-spin column was washed with 150 μl S-trap binding buffer by centrifugation at 4000 × g for 1 min. This process was repeated for a total of five washes. Twenty-five μl of 50 mM TEAB containing 1 μg trypsin was added to each sample, followed by proteolytic digestion for 2 h at 47 °C using a thermomixer (Eppendorf). Peptides were eluted with 50 mM TEAB pH 8.0 and centrifugation at 1000 × g for 1 min. Elution steps were repeated using 0.2% formic acid and 0.2% formic acid in 50% acetonitrile, respectively. The three eluates from each sample were combined and dried using a speed-vac before storage at −80 °C.

**Mass spectrometry.** Peptides were dissolved in 2% acetonitrile containing 0.1% formic acid, and each sample was independently analysed on an Orbitrap Exploris 480 mass spectrometer (Thermo Fisher Scientific), connected to an UltiMate 3000 RSLCnano System (Thermo Fisher Scientific). Peptides were injected on a PepMap 100 C18 LC trap column (300 μm ID × 5 mm, 5 μm, 100 Å) followed by separation on an EASY-Spray nanoLC C18 column (75 μm ID × 50 cm, 2 μm, 100 Å) at a flow rate of 250 nl/min. Solvent A was water containing 0.1% formic acid, and solvent B was 80% acetonitrile containing 0.1% formic acid. The gradient used for analysis was as follows: solvent B was maintained at 3% B for 5 min, followed by an increase from 3 to 35% B in 110 min, 35 to 50% B in 10 min, 50 to 90% B in 0.5 min, maintained at 90% B for 4 min, followed by a decrease to 3% in 0.5 min and equilibration at 2% for 20 min. The Orbitrap Exploris 480 was operated in positive-ion data-dependent mode. The precursor ion scan (full scan) was performed in the Orbitrap (OT) in the range of 400–1600 $m/z$ with a resolution of 120,000 at 200 $m/z$, an automatic gain control (AGC) target of $1 × 10^6$ and an ion injection time of 50 ms. MS/MS spectra were acquired in the OT using the Top 20 precursors fragmented by high-energy collisional dissociation (HCD) fragmentation. An HCD collision energy of 30% was used, the AGC target was set to $7.5 × 10^4$ and an ion injection time of 40 ms was allowed. Dynamic exclusion of ions within a ±10 ppm $m/z$ window was implemented using a 35 s exclusion duration. An electrospray voltage of 1.5 kV and capillary temperature of 280 °C, with no sheath and auxiliary gas flow, was used.

**Mass spectrometry data analysis.** All spectra were analysed using MaxQuant 1.6.10.43[66], and searched against the reviewed Homo sapiens Uniprot proteome database containing isoforms (downloaded on 11 January 2021) and the HA-GFP protein sequence (42,369 protein database entries in total). Peak list generation was performed within MaxQuant and searches were performed using default parameters and the built-in Andromeda search engine[67]. The enzyme specificity was set to consider fully tryptic peptides, and two missed cleavages were allowed. Oxidation of methionine, N-terminal acetylation and deamidation of asparagine and glutamine were allowed as variable modifications. Carbamidomethylation of cysteine was allowed as a fixed modification. A protein and peptide false discovery rate (FDR) of <1% was employed in MaxQuant. Proteins that contained similar peptides and that could not be differentiated on the basis of MS/MS analysis alone were grouped to satisfy the principles of parsimony. Reverse hits, contaminants and proteins only identified by site modifications were removed before downstream analysis. For label-free quantification, protein group intensities were extracted, median normalized, and filtered to contain at least three quantified values in one of the groups. Only proteins with at least two unique peptides were considered for relative quantification. Statistical testing was performed in the R statistical programming language (https://www.r-project.org/) using the LIMMA package[68] and the Benjamini-Hochberg correction for multiple hypothesis testing was implemented. Mitochondrial proteins were annotated based on annotations retrieved from Mitocarta 3.0[69]. The mass spectrometry proteomics data have been deposited to the

ProteomeXchange Consortium via the PRIDE partner repository[70] with the dataset identifier PXD026045.

**In vivo silica mouse model of membrane damage.** Eight to ten weeks-old, C57BL/6NJ female mice were exposed by intratracheal intubation with 40 µl aqueous suspensions of 200 µg silica crystals (MIN-U-SIL-15) or 200 µg red-fluorescent silica beads (control mice) in PBS. Mice were euthanised 16–18 h after instillation and bronchoalveolar lavage fluid was obtained by three consecutive instillations and withdrawals of 1 ml of 2 mM EDTA and 0.5% FCS in PBS. Recovered fluid was pelleted by centrifugation and treated with 300 µl of red blood lysis buffer for 5 min on ice. After that, cells were washed with 20 ml of PBS and centrifuged at $400 \times g$ for 7 min. Cells were then resuspended in 2 ml of PBS and counted. Cell viability was measured by trypan blue exclusion assay using a TC20 automated cell counter (Biorad). The percentage of viable cells was between 86 and 91%. Subsequently, cells were stained with Alexa Fluor 488 anti-mouse F4/80 antibody (Biolegend, 123120) and plated in µ-Slide 18 well coverslips (Ibidi, 81816) and processed for downstream applications.

**Single-cell generation, cDNA synthesis and library construction and sequencing protocol.** BAL samples from silica beads and silica-treated mice were treated for 10 min with red blood cell lysis solution (Miltenyi Biotec, 170080033) and washed with PBS by centrifugation at $300 \times g$ for 10 min. BAL cells were resuspended in PBS with 0.04 % BSA at a concentration of approximate 600 cells/µL for downstream processing. The quality and concentration of each single-cell suspension was measured using Trypan blue (Life tech) and the Eve automatic cell counter. Approximately 11,000 cells were loaded for each sample into a separate channel of a Chromium Chip G for use in the 10X Chromium Controller (cat: PN-1000120). The cells were partitioned into nanolitre scale Gel Beads in emulsions (GEMs) and lysed using the 10x Genomics Single Cell 3′ Chip V3.1 GEM, Library and Gel Bead Kit (cat: PN-1000121). cDNA synthesis and library construction were performed as per the manufacturer's instructions. The RNA was reversed transcribed and amplified using 14 cycles of PCR. Libraries were prepared from 10 µl of the cDNA and 12 cycles of amplification. Each library was prepared using Single Index Kit T Set A (cat: PN-1000213). Libraries were pooled according to estimated cell load, the pool was sequenced at 200pM on the HiSeq4000 system (Illumina) using the configuration 28-8-98 on a single-index-paired-end run to a depth of 332,315,673 reads and 330,550,363 reads for silica beads and crystals samples, respectively.

**Single-cell datasets analysis**
**Cluster annotation.** Data was processed using CellRanger v3.0.2 using the prebuilt mm10 index v3.0.0, and individual Seurat objects were created. Data was analysed with R v4.0.4 (The R Project for Statistical Computing). All the following functions belong to the Seurat v3.9.9 package[71], and their parameters are mentioned only if their value differ from the default one.

The following steps were applied to the data: (1) Cells with <500 features were excluded. (2) Samples were integrated using SCTransform workflow for pre-processing and normalization (2000 integration features were selected). (3) Cell cycle scoring was performed using the CellCycleScoring function, then the ScaleData function was used to subtract the cell cycle effect. (4) Principal component analysis was performed with the RunPCA function (npcs = 20). (5) Clustering was performed using the FindNeighbors function (dims = 1:20) and the FindClusters function (resolution = 0.8). (6) Dimension reduction was performed using RunUMAP (dims = 1:20). (7) Cluster annotation was done using the marker genes showed in the Supplementary Table 2 and following previously annotated or identified populations (Supplementary Table 2 and Supplementary Data 3). UMAP plots and expression dot plots were made using respectively the DimPlot and DotPlot (with the RNA assay) functions (both from Seurat).

**Differential expression analysis.** Differential expression analysis for each cluster between the two conditions (Silica beads and Silica crystals) was performed using the FindMarkers function of Seurat (DESeq2 test). The Seurat DESeq2DETest function was modified to output the Wald statistic. Only genes with an adjusted $p$-value lower than 0.05 between two comparisons were considered statistically significant.

**Gene Set Enrichment Analysis.** Gene Set Enrichment Analysis (GSEA)[72] was performed with R v4.0.5 (The R Project for Statistical Computing) using the fgsea v1.16.0 package[73]. The genes were ranked according to the Wald statistic resulting from the differential expression analysis, and GSEA was run for the C2-CP:REACTOME v7.4 geneset from the MSigDB collection[74], with 1,000,000 permutations and the standard score type. Only pathways with an adjusted $p$-value lower than 0.05 were considered statistically significant. Dot plots illustrating selected pathways were made with ggplot2 (The R Project for Statistical Computing).

**Statistical analysis**
Statistical analysis was performed using GraphPad Prism 10 software or R 3.6.3 (The R Project for Statistical Computing). High-content imaging analysis and mean values were obtained using R 3.6.3. The number of biological replicates and the statistical analysis performed, and post hoc tests used can be found in the figure legends. The statistical significance of data is denoted on graphs by asterisks (*) where (*) = $p < 0.05$, (**) = $p < 0.01$, (***) = $p < 0.001$ or ns = not significant.

Plotting: All graphs were plotted in GraphPad Prism software, with the exception of the graphs showed in Extended Fig. 3b and sc-RNA-seq graphs that were plotted using R 3.6.3. Figures schemes were created with BioRender.com.

**Reporting summary**
Further information on research design is available in the Nature Portfolio Reporting Summary linked to this article.

## Data availability
The scRNA-seq data generated in this study have been deposited at GEO (accession number GSE174414) and can be accessed through the website https://shiny.crick.ac.uk/033_scrnaseq_airspace_cells_inflammation/865eb86c8eca0/ The mass spectrometry proteomics data have been deposited to the ProteomeXchange Consortium via the PRIDE partner repository with the dataset identifier PXD026045". Metabolomics data have been deposited in Zenodo, under the following https://doi.org/10.5281/zenodo.5495849. All other data needed to evaluate the conclusions of the study are present in the paper or in the supplementary materials. Source data are provided with this paper.

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

## Acknowledgements

We thank Sharon Tooze, Jeremy Carlton, Nicholas Ktistakis and Venizelos Papayannopoulos for critical reading the manuscript and Prof. Anne Simonsen for the NIPSNAP vectors. We thank Dr. Varda Shoshan-Barmatz for sharing protocols to evaluate VDAC oligomerization. We are also grateful to the Human Embryonic Stem Cell Unit, Advanced Light Microscopy, Electron Microscopy, Advanced Sequencing facilities and Bioinformatics analysis at the Crick for their support in various aspects of the work. We thank Jimena Perez Lloret for scRNA-seq sample preparation. This work was supported by the Francis Crick Institute (to M.G.G.), which receives its core funding from Cancer Research UK (FC001092), the UK Medical Research Council (FC001092), and the Wellcome Trust (FC001092). This project has received funding from the European Research Council (ERC) under the European Union's Horizon 2020 research and innovation programme (grant agreement n° 772022). C.B. has received funding from the European Respiratory Society and the European Union's H2020 research and innovation programme under the Marie Sklodowska-Curie grant agreement No 713406. P.S. is the recipient of a H2020 MSCA Individual Fellowship (project SpaTime_AnTB n°892859). T.R. has received funding from the Deutsche Forschungsgemeinschaft – DFG - SFB 850 Project B7, G.R.K. 2606 (Project ID 423813989) and the German Consortium for Translational Cancer Research (DKTK). For the purpose of Open Access, the author has applied a CC BY public copyright licence to any Author Accepted Manuscript version arising from this submission.

## Author contributions

Conceptualization: M.G.G. and C.B. Formal analysis: C.B., T.H., N.B. and M.S.D.S. Funding acquisition: M.G.G. Investigation: M.G.G. and C.B. Methodology: C.B., T.H., E.P., E.M.B., N.B., M.S.D.S., P.S., B.A., A.R., A.F., J.M., C.M., J.I.M., M.G., T.R., M.T. and M.G.G. Project administration: M.G.G. and C.B.. Supervision: M.G.G., J.I.M. and M.T. Visualization: C.B. and N.B. Writing—original draft: M.G.G. and C.B. Writing—review & editing: C.B., T.H., E.P., E.M.B., N.B., M.S.D.S., P.S., B.A., A.R., A.F., C.M., J.I.M., M.G., T.R., M.T. and M.G.G.

## Funding

## Competing interests

The authors declare no competing interests.
