## [Peer Review File · Nature Communications]

Lysosomal damage drives mitochondrial proteome remodelling and reprograms macrophage immunometabolismEditorial Note: This manuscript has been previously reviewed at another journal that is not operating a transparent peer review scheme. This document only contains reviewer comments and rebuttal letters for versions considered at *Nature Communications*.

REVIEWERS' COMMENTS

Reviewer #1 (Remarks to the Author):

The authors report that lysosomal damage upon *M. tuberculosis* infection causes lysosomal leakage and metabolic reprogramming of mitochondria in macrophages. Several interesting observations are reported, but the most critical point has been the proposal of the authors that lysosomal cathepsins are imported into mitochondria to degrade mitochondrial proteins. Although in my opinion some doubts are justified that the proposed mechanism is indeed correct, the authors use now more careful wording when interpreting their results. Future experiments will clarify the exact mechanism of mitochondrial reprogramming in response to lysosomal leakage.

Reviewer #2 (Remarks to the Author):

This is a revised version of a manuscript I previously reviewed for Nature Cell Biology (reviewer #5). I was not involved in the first rounds of review of the previous submission, but was only later asked by the NCB editor to specifically comment on the current evidence provided in the paper to support the authors' claim that lysosomal proteases enter mitochondria. Unfortunately, I cannot find any changes to the manuscript in response to my concerns, all of which therefore remain valid. One of my major points is that the data presented by the authors are insufficient to support their claim that lysosomal hydrolases enter the mitochondrial matrix. For example, in relation to Figure 4a the authors state in their rebuttal letter: "...we are not stating that the particles in the cytosol/OMM are inside the mitochondria". However, in the text, lines 203-205, they continue to say: "Strikingly, after LLOMe treatment, gold particles were present throughout the cytosol and found inside mitochondria (Fig. 4a,b) strongly suggesting that lysosomal contents are transferred to mitochondria after damage." As outlined in my previous review, I don't even believe that the particles that are labelled "matrix" in the modified figure provided in the rebuttal letter are actually inside the matrix. Therefore, simply changing the labelling of the figure is not sufficient to resolve this issue. The authors have not even corrected the incorrect statement in the legend of Figure 4a that the gold particles are 15 nm in size. Since the authors seem unwilling to improve their manuscript based on these (and other) comments, I recommend to finally reject it.

Point-by-point reply to reviewers

Reviewer #1

The authors report that lysosomal damage upon *M. tuberculosis* infection causes lysosomal leakage and metabolic reprogramming of mitochondria in macrophages. Several interesting observations are reported, but the most critical point has been the proposal of the authors that lysosomal cathepsins are imported into mitochondria to degrade mitochondrial proteins. Although in my opinion some doubts are justified that the proposed mechanism is indeed correct, the authors use now more careful wording when interpreting their results. Future experiments will clarify the exact mechanism of mitochondrial reprogramming in response to lysosomal leakage.

We thank the reviewer for the valuable feedback.

Reviewer #2

This is a revised version of a manuscript I previously reviewed for *Nature Cell Biology* (reviewer #5). I was not involved in the first rounds of review of the previous submission, but was only later asked by the NCB editor to specifically comment on the current evidence provided in the paper to support the authors' claim that lysosomal proteases enter mitochondria. Unfortunately, I cannot find any changes to the manuscript in response to my concerns, all of which therefore remain valid. One of my major points is that the data presented by the authors are insufficient to support their claim that lysosomal hydrolases enter the mitochondrial matrix. For example, in relation to Figure 4a the authors state in their rebuttal letter: "...we are not stating that the particles in the cytosol/OMM are inside the mitochondria". However, in the text, lines 203-205, they continue to say: "Strikingly, after LLOMe treatment, gold particles were present throughout the cytosol and found inside mitochondria (Fig. 4a,b) strongly suggesting that lysosomal contents are transferred to mitochondria after damage." As outlined in my previous review, I don't even believe that the particles that are labelled "matrix" in the modified figure provided in the rebuttal letter are actually inside the matrix. Therefore, simply changing the labelling of the figure is not sufficient to resolve this issue. The authors have not even corrected the incorrect statement in the legend of Figure 4a that the gold particles are 15 nm in size. Since the authors seem unwilling to improve their manuscript based on these (and other) comments, I recommend to finally reject it.

Following the suggestions of the reviewer, we have now moved the figure 4a to the supplementary and changed/corrected the text as recommended.